# The antifungal caspofungin increases fluoroquinolone activity against *Staphylococcus aureus* biofilms by inhibiting *N*-acetylglucosamine transferase

Wafi Siala[1], Soňa Kucharíková[2,3], Annabel Braem[4], Jef Vleugels[4], Paul M. Tulkens[1],
Marie-Paule Mingeot-Leclercq[1], Patrick Van Dijck [2,3] & Françoise Van Bambeke[1]

Biofilms play a major role in *Staphylococcus aureus* pathogenicity but respond poorly to antibiotics. Here, we show that the antifungal caspofungin improves the activity of fluoroquinolones (moxifloxacin, delafloxacin) against *S. aureus* biofilms grown *in vitro* (96-well plates or catheters) and *in vivo* (murine model of implanted catheters). The degree of synergy among different clinical isolates is inversely proportional to the expression level of *ica* operon, the products of which synthesize poly-*N*-acetyl-glucosamine polymers, a major constituent of biofilm matrix. *In vitro,* caspofungin inhibits the activity of IcaA, which shares homology with β-1-3-glucan synthase (caspofungin's pharmacological target in fungi). This inhibition destructures the matrix, reduces the concentration and polymerization of exopolysaccharides in biofilms, and increases fluoroquinolone penetration inside biofilms. Our study identifies a bacterial target for caspofungin and indicates that IcaA inhibitors could potentially be useful in the treatment of biofilm-related infections.

[1] Pharmacologie cellulaire et moléculaire, Louvain Drug Research Institute, Université catholique de Louvain, 1200 Brussels, Belgium. [2] Laboratory of Molecular Cell Biology, Institute of Botany and Microbiology, KULeuven, 3000 Leuven, Belgium. [3] Department of Molecular Microbiology, VIB, KULeuven, 3000 Leuven, Belgium. [4] Department of Materials Engineering, KULeuven, 3000 Leuven, Belgium. Correspondence and requests for materials should be addressed to F.V.B. (email: francoise.vanbambeke@uclouvain.be).

Staphylococcus aureus is one of the most prevalent human pathogens in the Western world, being capable of causing a wide spectrum of community- or hospital-acquired infections. S. aureus healthcare-associated infections are related to the capacity of this bacterium to form biofilms[1]. These consist of complex communities of microorganisms encased in a glycocalyx composed of DNA, proteins and polysaccharides. Biofilms not only contribute to bacterial colonization of surfaces but also represent a reservoir for continuing bacterial dissemination within the body. Thus, staphylococcal biofilms are considered as a main reason for persistence and/or recurrence of infections like endocarditis, osteomyelitis or those associated with indwelling medical devices[2,3]. These infections are also prone to treatment failure[4], ascribed to poor bacterial response to immune defenses and antibiotics[5–7]. Unresponsiveness to antibiotics is related to the facts that (i) biofilm matrix opposes a barrier to the access of host defenses and antibiotics to embedded bacteria, and (ii) bacteria within biofilms adopt a dormant lifestyle poorly responsive to antibiotic action[8]. Antibiotic combination has been considered as a valuable strategy to act on staphylococcal biofilms[9,10], but this approach does not address the main pharmacokinetic issue posed by biofilms, consisting in insufficient drug penetration within the structure.

In strains of S. aureus expressing the ica operon, a major constituent of the biofilm matrix is poly-N-acetyl-glucosamine (PNAG) polymer, also referred to as polysaccharide intercellular adhesin (PIA)[5,11,12]. The gene products of the icaADBC locus include IcaA (transmembrane N-acetyl-glucosamine transferase synthesizing short PNAG polymers[13]), IcaD (protein increasing the biosynthetic efficiency of IcaA and playing a predominant role in the synthesis of oligomers longer than 20 residues[13]), IcaB (extracellular N-deacetylase enabling PNAG fixation at the bacterial cell surface and biofilm formation[1,14]), and IcaC (putative transmembrane protein initially considered as involved in the polymerization of short chain polymers[13] but more recently, being recognized as a O-succinyltransferase catalyzing the O-modification of PNAG during biosynthesis[15]). Expression of icaA and subsequent PNAG production have been associated with the capacity of S. aureus to produce biofilm in vitro, including for clinical isolates collected from device-related infections[16–18]. The expression of the icaADBC

locus in S. aureus depends on the genetic background of the strain and is upregulated in vivo[19]. Moreover, PNAG-enriched biofilms are effectively dispersed by the glycoside hydrolase dispersin B, positioning this polysaccharide as an attractive target for adjunctive therapy[20,21]. Yet, the applicability of dispersin B itself in the clinics is limited to the field of wound or catheter-related infections by its proteic nature[22,23].

Several alternative, non-protein-based strategies have thus been proposed to improve antibiotic activity against staphylococcal biofilms[24]. Small molecules like quinolines[25], 2-aminobenzimidazoles[26] or norspermidine and guanidine or biguanide biomimetics[27] have proven effective in vitro but have never been tested in vivo, so that their druggability is unknown. Moreover, their mechanism of action is only partially elucidated, making a successful lead optimization and development of more potent analogues uncertain.

We set out to identify, amongst already approved drugs, compounds that would act on extracellular matrix to increase antibiotic activity against staphylococcal biofilms. This was thought to facilitate the potential future clinical exploitation of the results. On the basis of the importance of polysaccharidic compounds in the matrix of staphylococcal biofilms, we selected for this study caspofungin, an approved antifungal echinochandin[28], which acts on Candida and Aspergillus species by inhibiting β-1-3-glucan synthase[29]. We used clinical isolates of S. aureus previously demonstrated to be recalcitrant to the action of antibiotics when grown as biofilms[30]. We compared two fluoroquinolone antibiotics, namely, (a) moxifloxacin, considered as the most potent anti-Gram-positive fluoroquinolone among those available on the market[31], but which is only modestly active against biofilms[32], and (b) delafloxacin, an even more potent anti-Gram-positive fluoroquinolone currently in phase III of clinical development[33], which also showed more promising activity than moxifloxacin against biofilms[30].

We demonstrate that caspofungin markedly improves the activity of both fluoroquinolones in in vitro and in vivo models of biofilms. This synergy is due to the capacity of caspofungin to inhibit the enzymatic activity of IcaA, which shares homology with the fungal β-1-3-glucan synthase. Thus, we establish a bacterial target for this class of antifungal compounds and document a therapeutic potential of pharmacological inhibitors of IcaA.

**Table 1 | Activity of moxifloxacin and delafloxacin alone or combined with caspofungin against planktonic bacteria and 24-h-biofilms.**

| Strain* | MIC† (mg l⁻¹) | | | Fluoroquinolone concentrations (mg l⁻¹) needed to reduce bacterial viability in biofilms of 50%‡ | | | |
|---|---|---|---|---|---|---|---|
| | MXF† | DFX† | CAS† | MXF | | DFX | |
| | | | | Alone | + CAS§ | Alone | + CAS§ |
| ATCC33591 (MRSA) | 0.032 | 0.004 | 80 | 1.25 | 0.1 | 0.125 | 0.125 |
| 2011S027 (MSSA) | 0.125 | 0.004 | 80 | 0.9 | 0.7 | 0.5 | 0.125 |
| Surv2003/1083 (MSSA) | 0.125 | 0.004 | 160 | >20‖ | 17 | >20‖ | 2 |
| 2009S025 (MRSA) | 0.125 | 0.125 | 80 | >20 | 1.9 | 4 | 1 |
| Surv2005/104 (MRSA) | 2 | 0.125 | 160 | >20 | 18 | >20 | 2 |
| Surv2005/179 (MRSA) | 2 | 0.016 | 80 | >20 | 3.8 | >20 | 4 |
| 2009S028 (MRSA) | 2 | 0.016 | 80 | >20 | 3.7 | 8 | 0.5 |
| Surv2003/651 (MRSA) | 2 | 0.125 | 160 | >20 | >20 | >20 | 8 |
| Xen36 (MSSA) (bioluminescent strain derived from S. aureus ATCC 49525) | 1 | 0.016 | 80 | >20 | 10.4 | >20 | 1 |

*All clinical isolates belong to the epidemic CC5 or CC8 clonal complexes; see Siala et al.[30] for origin and description; MSSA, methicillin susceptible S. aureus; MRSA, methicillin-resistant S. aureus.
†MIC, minimal inhibitory concentration; MXF, moxifloxacin; DFX, delafloxacin; CAS, caspofungin.
‡Calculated using the Hill function fitted to the data of concentration–response experiments similar to those presented in Supplementary Fig. 3 for selected strains.
§Used at 40 mg l⁻¹.
‖ Effect not reached at 20 mg l⁻¹.

## Results

**Caspofungin-fluoroquinolone activity on biofilms *in vitro*.** In a first set of experiments, we examined the activity of moxifloxacin and delafloxacin alone or combined with caspofungin at a fixed concentration (40 mg l$^{-1}$) against *S. aureus* mature biofilms grown in 96-well plates. The laboratory strain ATCC33591 and seven clinical strains, previously described as clinical isolates forming biofilms *in vitro*[30], were used in parallel (see Table 1 for minimal inhibitory concentrations (MICs)). We first checked that caspofungin did not affect the bacterial viability or biomass in biofilms when used alone at the fixed concentration selected (Supplementary Fig. 1). Figure 1 illustrates typical results for 4

biofilms exposed during 48 h to fluoroquinolones alone or combined with caspofungin (data for the other four strains under study are shown in Supplementary Fig. 2). In a first step, we examined the effect of increasing concentrations of fluoroquinolones on bacterial viability in biofilms, as assessed in parallel by the measure of residual resorufin fluorescence (bacterial metabolic activity; left axis) and of colony-forming units (CFUs) (viable bacteria; right axis). Moxifloxacin alone (Fig. 1a–d) was poorly active on these biofilms, reaching a bactericidal effect (3 log$_{10}$ decrease in CFUs) only against ATCC33591 biofilm at the highest concentration tested (shown in Supplementary Fig. 2). Delafloxacin alone (Fig. 1i–l) reached a

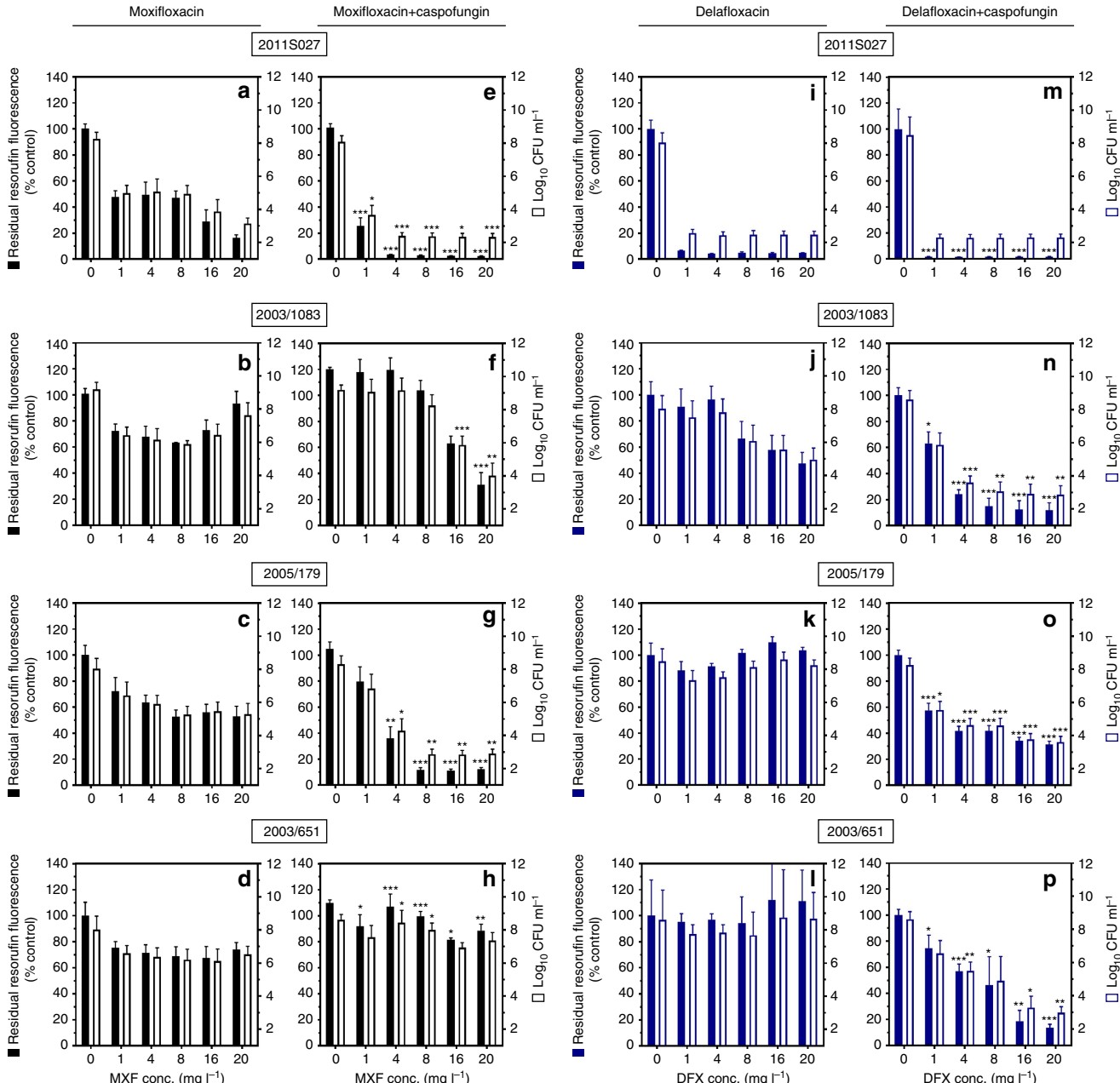

**Figure 1 | Effect of fluoroquinolones alone or combined with caspofungin on biofilms from four selected strains.** Biofilms were incubated during 48 h in the absence or in the presence of the drugs (moxifloxacin (**a–h** black) or delafloxacin (**i–p** blue) at increasing concentrations and used alone (**a–d** and **i–l**) or combined (**e–h** and **m–p**) with 40 mg l$^{-1}$ caspofungin). The ordinate shows resorufin fluorescence (closed bars; left scale; expressed in percentage of the value measured in control conditions (no fluoroquinolone added)) and CFU (open bars; right scale; expressed in log$_{10}$ units). Data are the mean ± s.d. of four replicates. Statistical analysis: multiple *t*-tests comparing data for fluoroquinolone alone or combined with caspofungin in the same conditions: ***P < 0.001; **P < 0.01; *P < 0.05.

bactericidal effect against the ATCC33591 (Supplementary Fig. 2) and 2011S027 but not against the other strains. In sharp contrast, when fluoroquinolones were combined with caspofungin, a bactericidal effect was observed against 3 out of 4 strains for moxifloxacin (Fig. 1e–h) and against all strains for delafloxacin (Fig. 1m–p), for which this effect was also reached at lower concentrations ($\leq 4$ mg l$^{-1}$). A similar improvement of activity was observed when combining caspofungin with fluoroquinolones against biofilms from the other strains under study (Supplementary Fig. 2).

Because the same type of results was obtained when determining residual viability based on CFU counts or resorufin fluorescence, we used the latter technique to obtain full concentration–response curves, which allowed us to determine and compare the relative potencies of the drugs (that is, the concentrations needed to reach a specified effect) against these biofilms. We also evaluated in the same conditions the effect of fluoroquinolones alone or combined with caspofungin on biofilm biomass using crystal violet staining (Supplementary Fig. 3 for an illustration for four selected strains). The Hill function fitted to the data of these concentration–response curves was used to calculate the concentrations of each fluoroquinolone (used alone or combined with 40 mg l$^{-1}$ caspofungin) needed to reduce bacterial viability of 25, 50 or 75% within biofilms compared with control, and the corresponding potencies are shown graphically in Fig. 2 for the eight strains investigated. Moxifloxacin (Fig. 2a) alone only reduced viability of 25% in six strains and of 50% in two strains. In contrast, its combination with caspofungin (40 mg l$^{-1}$) achieved 75% reduction of viability for seven strains, with only 2003/651 remaining unaffected by this treatment (Fig. 2b). For delafloxacin alone (Fig. 2c), a 50 and 75% reduction of viability was obtained for four and three strains, respectively, in the range of concentrations investigated, and the corresponding potencies were increased (lower values) when combined with caspofungin (Fig. 2d). For two strains, however, a 75% reduction in viability could not be achieved in the range of concentrations tested even in combination with caspofungin (maximum reduction observed: 65% for 2005/179 and 68% for 2009S028, respectively). Considering then drug effects on biomass, a reduction in crystal violet staining was observed (although to a lesser extent than for viability) for all strains when caspofungin was combined with delafloxacin but only for two of them (2011S027 and 2005/179) when it was combined with moxifloxacin.

Using the same experimental design, we also tested the effect of caspofungin on the activity of three other widely used antistaphylococal agents, namely vancomycin, daptomycin and linezolid (Supplementary Table 1). Synergy with caspofungin was observed with no strain when combined with vancomycin, for only one strain when combined with daptomycin, and for only four out of eight strains when combined with linezolid.

The activity of fluoroquinolones and caspofungin combined at fixed concentrations (10 and 40 mg l$^{-1}$, respectively) was then examined in a second *in vitro* model consisting in biofilms formed inside polyurethane catheter pieces with the seven clinical strains examined so far and with the bioluminescent strain Xen36 (Fig. 3). When tested alone, caspofungin and moxifloxacin were ineffective in this model while delafloxacin significantly reduced bacterial counts for all strains except 2003/651 (with residual counts remaining, however, $\geq 4.5$ log$_{10}$ CFUs for four strains). When used in combination, a marked synergy between each fluoroquinolone and caspofungin was observed. Thus, moxifloxacin gained considerable activity against all strains except 2003/651 and delafloxacin activity was improved against five strains, including 2003/651. While the extent of synergy widely differed between strains (with reduction in CFU varying

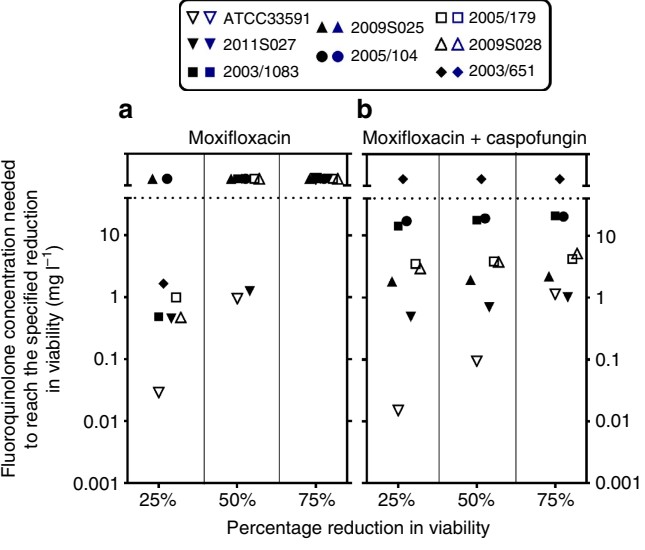

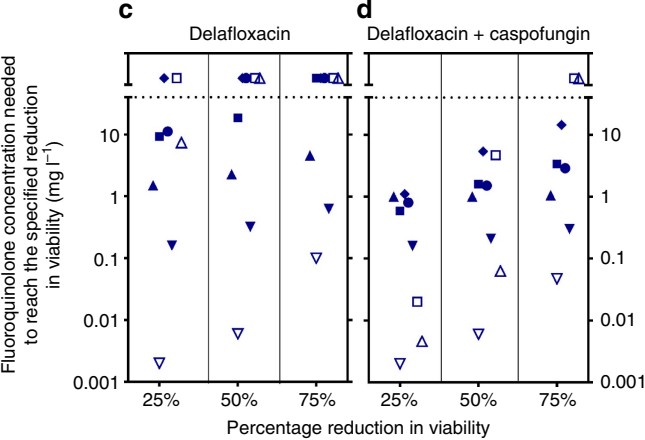

**Figure 2 | Relative potencies of fluoroquinolones alone or combined with caspofungin on biofilms.** Comparison of relative potencies of fluoroquinolones alone (**a** moxifloxacin; **c** delafloxacin) or combined with a fixed concentration (40 mg l$^{-1}$) of caspofungin (**b** and **d**) against biofilms. The ordinate shows the concentrations of fluoroquinolones needed to reach 25, 50 or 75% reductions in viability as assessed by measuring residual resorufin fluorescence. Active concentrations were calculated based on the equation of sigmoid concentration–response curves obtained for each strain in experiments similar to those illustrated in Supplementary Fig. 3. Each symbol corresponds to a specific strain, as indicated on the top of the graphs. A lower active concentrations corresponds to a higher potency. The horizontal dotted lines separate values for which calculated concentrations were above the actual maximal fluoroquinolone concentrations tested (20 mg l$^{-1}$).

for moxifloxacin between 1.9 and 7.6 log$_{10}$ for strains 2003/651 and 2011S027, respectively), it was more marked for strains showing more adhesion to the catheters (2011S027 and 2003/1083).

**Caspofungin-fluoroquinolone activity on biofilms *in vivo*.** In a next step, we assessed whether fluoroquinolones, caspofungin, or their combination could act *in vivo* on *S. aureus* biofilms present on catheters. Biofilms were first made *in vitro* and the infected catheters implanted under the skin of BALB/c mice. Biofilms were then allowed to develop *in vivo* for 24 h, after which animals were treated twice daily with either 40 mg kg$^{-1}$ of fluoroquinolone alone or once daily with 4 mg kg$^{-1}$ of caspofungin alone or with

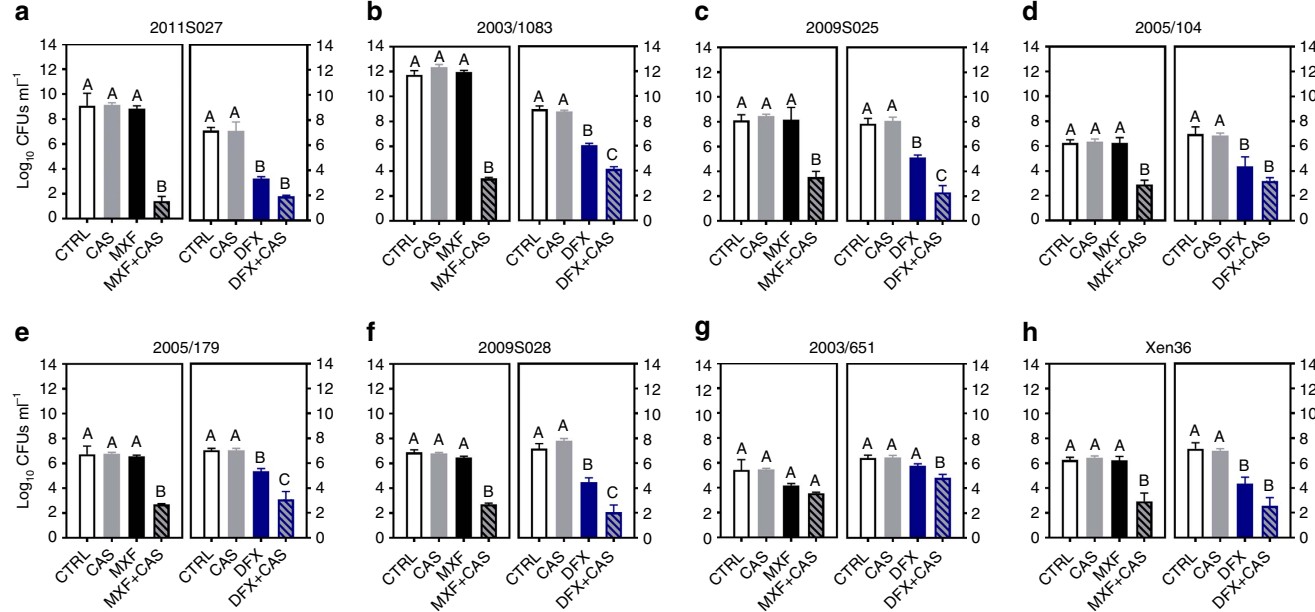

**Figure 3 | Effect of caspofungin fluoroquinolones used alone or in combination on biofilms grown in catheters *in vitro*.** Biofilms grown in catheters for clinical isolates (**a–g**) and the bioluminescent strain Xen36 (**h**) during 24 h were exposed to caspofungin (CAS; 40 mg l$^{-1}$), fluoroquinolones alone (10 mg l$^{-1}$) or combined with 40 mg l$^{-1}$ CAS. In each graph, the left panel shows data obtained with moxifloxacin (MXF; black) and the right panel, data obtained with delafloxacin (DFX; blue) and matching controls for each strain. The graphs show the number of CFU (in log$_{10}$ units) recovered from catheters after 48 h of incubation in the absence (control (CTRL)) or in the presence of the drugs. Data are means ± s.d. for three catheters. Statistical analysis: bars with different letters show data that significantly different from one another ($P < 0.01$; one-way ANOVA with Tukey *post-hoc* test).

their combination for 7 days. These doses and schedules were selected as mimicking human exposure in clinical practice taking into consideration the shorter half-life of moxifloxacin in mice compared with humans[34–37]. For delafloxacin, in the absence of published humanized pharmacokinetic data, we selected as a starting point a dose equivalent to that of moxifloxacin, taking also into account its short half-live and high protein binding in mice[38]. In a first experiment, we followed on a daily basis mice with Xen36-infected catheters using bioluminescence imaging (Fig. 4a). In untreated mice, the intensity of the bioluminescence signal increased almost linearly from day 1 to 4 and reached thereafter a plateau (Fig. 4b,c). No difference in signal intensity was observed between untreated mice and those treated with caspofungin alone. Moxifloxacin alone was also ineffective over the whole treatment duration (Fig. 4b), but delafloxacin caused a marked decrease in bioluminescence signal as from day 1 (Fig. 4c). When moxifloxacin was combined with caspofungin, the bioluminescence signal was significantly lower from day 4 as compared with mice treated by moxifloxacin alone or caspofungin alone (Fig. 4b). When delafloxacin was combined with caspofungin, no significant difference was observed with animals treated by delafloxacin alone if considering the mean values for all catheters (Fig. 4c). Yet, the establishment of the infection was slower in part of the mice and full eradication was achieved in one of the mice treated by the combination (compare Fig. 4e and Fig. 4d).

The experiment was extended in the same conditions to catheters infected by two clinical isolates against which the combinations were respectively markedly (2011S027) or marginally (2003/651) more effective *in vitro* than for fluoroquinolones alone (Fig. 3). CFUs remaining on catheters were counted at day 7. As shown in Fig. 5a–f, caspofungin alone was ineffective against all biofilms. Fluoroquinolones alone caused a limited (∼1 log$_{10}$ for moxifloxacin) or a marked (4.5–7 log$_{10}$ for delafloxacin) decrease in the number of CFUs recovered from the catheters infected by 2011S027 and Xen36 but not by 2003/651.

In contrast, the number of CFUs recovered from the catheters was significantly lower in animals treated with the combination of moxifloxacin and caspofungin (mean reduction of 2.1, 1.6 and 0.4 log$_{10}$ CFUs for strains 2011S027, Xen36 and 2003/651, respectively (Fig. 5a–c)) or with the combination of delafloxacin and caspofungin for strain 2003/651 (reduction of 0.5 log$_{10}$ CFUs (Fig. 5f)). Against the two other strains, combining caspofungin with delafloxacin allowed to achieve total (2011S027) or partial (Xen36) sterilization but the difference was not significant with the already impressive effect reached in mice treated with delafloxacin alone (Fig. 5d,e). To better apprehend the potential of combining delafloxacin with caspofungin, we therefore performed a dose–response study, using the highly responsive strain 2011S027 (Fig. 5g). Mice were treated with delafloxacin at increasing doses of 10, 20 or 40 mg kg$^{-1}$ twice daily alone or combined with caspofungin 4 mg kg$^{-1}$ once daily for 7 days and catheter-associated CFUs were determined at the end of this treatment. Delafloxacin activity was clearly dose-dependent over this range, and the synergy with caspofungin was best seen at the lowest dose (10 mg kg$^{-1}$) of delafloxacin, which was suboptimal in monotherapy.

**Electron microscopy studies of *in vivo* biofilms.** Catheters with biofilms made by the clinical isolate 2011S027 and recovered from mice after *in vivo* treatment with moxifloxacin alone, caspofungin alone or their combination, that is, focusing on conditions in which the effect of the combination was most evident, were examined by scanning electron microscopy (Fig. 6). A massive biofilm matrix with only a few visible bacterial cells was observed on the surface of catheters extracted from control (saline-treated) (Fig. 6a), caspofungin- (Fig. 6b) or moxifloxacin-treated animals (Fig. 6c). Some cracks (possibly due to the drying process) did, however, appear on caspofungin-treated biofilms. More strikingly, catheters extracted from animals treated with the combination of moxifloxacin and caspofungin (Fig. 6d) showed a

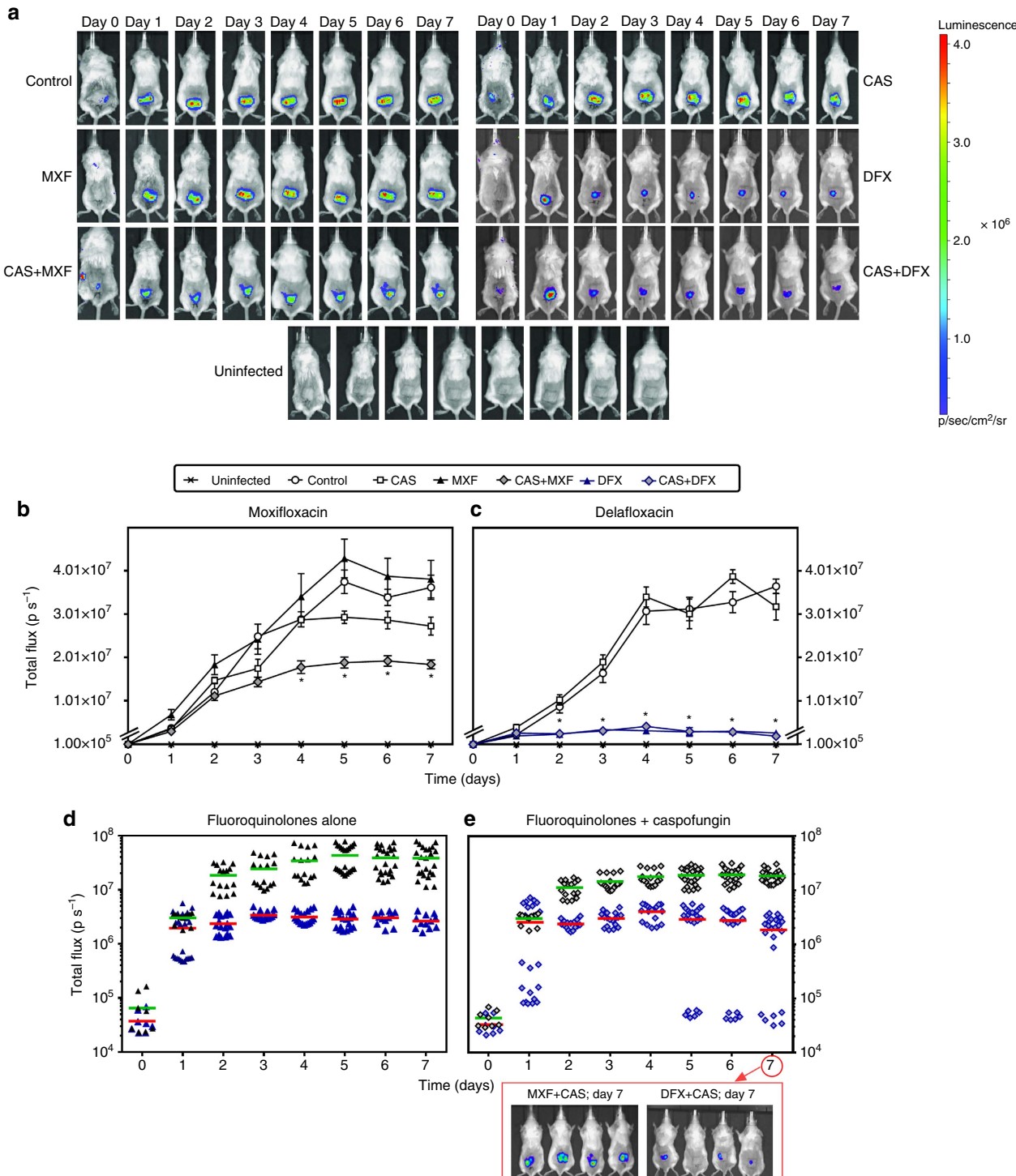

**Figure 4 | Activity of fluoroquinolones alone or combined with caspofungin against the bioluminescent strain Xen36 *in vivo*.** Bioluminescent signal emitted from catheters infected by Xen36, implanted at day 0 in the back of mice treated 24 h after implantation and for the next 7 days with caspofungin (CAS) (4 mg kg$^{-1}$ of body weight once daily), moxifloxacin (MXF) or delafloxacin (DFX) (40 mg kg$^{-1}$ of body weight twice daily) or with a fluoroquinolone and caspofungin (each injected separately and according to its own schedule; (CAS + MXF) or (CAS + DFX)). All drugs were given by intraperitoneal injection. Control: animals implanted with infected catheters and treated by normal saline (0.9% NaCl) twice daily. Uninfected: animals implanted with non-infected catheters and left untreated (used for detection of background signal). (**a**) Representative bioluminescence images for one mouse per group: intensity of the transcutaneous photon emission represented as a pseudocolor image. (**b,c**) Quantitative analysis per fluoroquinolone (moxifloxacin and matching controls (**b**); delafloxacin and matching controls (**c**)): *in vivo* bioluminescence signals expressed in photons per second (p s$^{-1}$), with data expressed as means ± s.e.m. Statistical analysis: two-way ANOVA, Tukey *post-hoc* test. *$P < 0.001$ when comparing combinations versus fluoroquinolones alone. (**d,e**) Quantitative analysis comparing both fluoroquinolones when given alone (**d**) or in combination with caspofungin (**e**). The data are shown as individual values with the corresponding means represented by a horizontal coloured line. Images are those of four mice treated during 7 days by the combinations.

destroyed biofilm structure, with less visible matrix and an abundance of bacterial cells in patches spread on the surface.

**Caspofungin effect on fluoroquinolone penetration in biofilms.** Previous studies have documented a correlation between the activity of antibiotics against biofilm-encased bacteria and their capacity to penetrate the biofilm[30,39,40]. Taking advantage of the intrinsic fluorescence of fluoroquinolones, we examined their

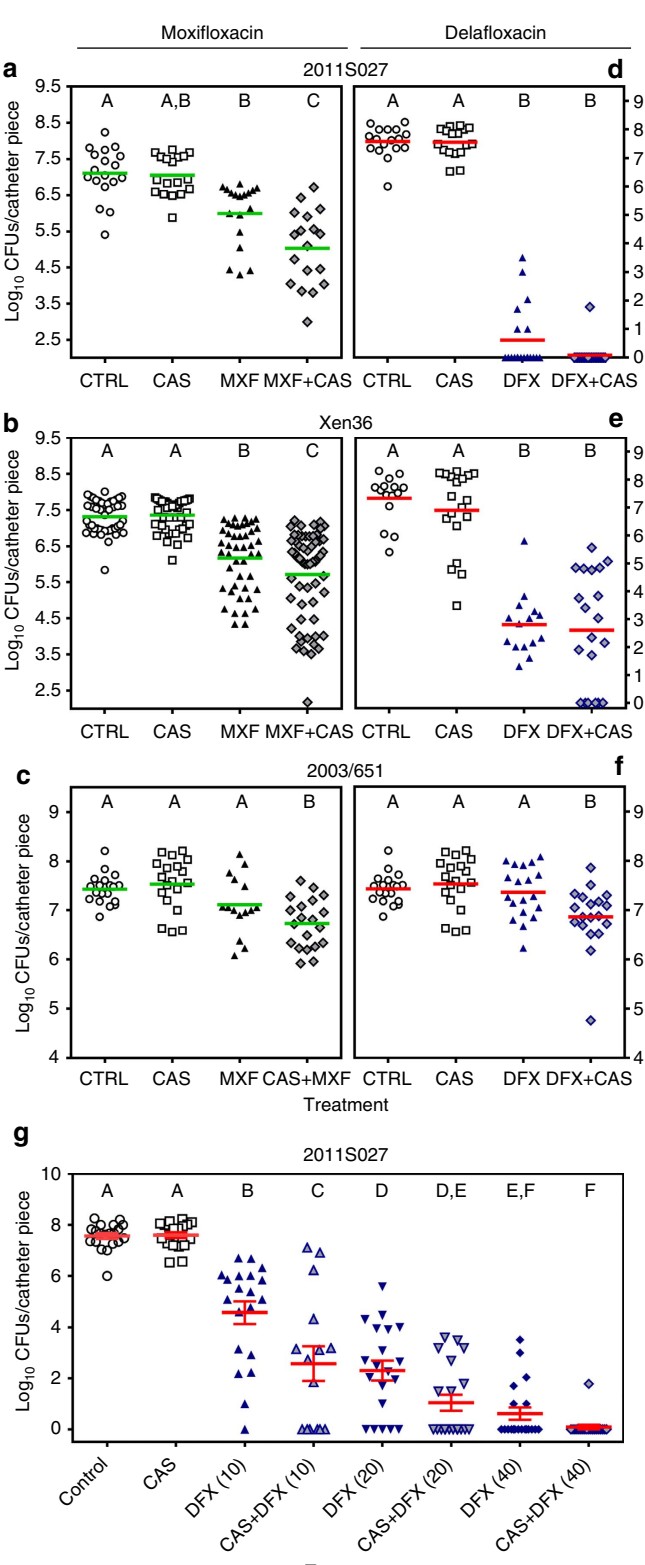

penetration within biofilms grown on glass cover slips using confocal laser scanning microscopy. Figure 7 shows the concentration of fluoroquinolones in the depth of biofilms produced by four clinical isolates and exposed to $20\,mg\,l^{-1}$ of antibiotics alone or combined with $40\,mg\,l^{-1}$ caspofungin, together with the corresponding microscopic images. While fluoroquinolone penetration was important for the biofilm produced by strain 2011S027 (Fig. 7a,e), it was minimal for the three other strains. Combination with caspofungin largely increased fluoroquinolone penetration not only in biofilm produced by 2011S027 but also in those produced by 2003/1083 and 2005/179 (Fig. 7a–c, e–g). It also increased delafloxacin (Fig. 7h) but not moxifloxacin (Fig. 7d) penetration in biofilms produced by 2003/651.

**Caspofungin effects on biofilm matrix properties.** Because poly-$\beta$(1-6)-$N$-acetylglucosamine is a major matrix component[11], we examined whether the enhancement of fluoroquinolone penetration and the destructuration of biofilm matrix induced by caspofungin could be related to a modification in the concentration or polymerization degree of this exopolysaccharide. To this effect, the abundance of poly-$\beta$(1-6)-$N$-acetylglucosamine in biofilms was compared in control conditions and after incubation with caspofungin, for the same four clinical isolates and the reference strain ATCC33591, using an anti-PNAG antiserum (Fig. 8a). Caspofungin markedly decreased the signal for all strains, except 2003/651, suggesting it could interfere with the metabolism of this polysaccharide.

A critical property of poly-$\beta$(1-6)-$N$-acetylglucosamine as a matrix constituent is its degree of polymerization, since hydrolysis of polymers leads to biofilm dispersal[20]. The influence of caspofungin on the degree of PNAG polymerization was therefore evaluated in biofilms from the same strains after 24 h of culture in the absence or in the presence of $40\,mg\,l^{-1}$ caspofungin. To this effect, PNAG were purified from these biofilms and submitted to a treatment by dispersin B to release $N$-acetylglucosamine monomers and allow for their quantification[20] (Fig. 8b). In control conditions, more monomers were generated by dispersin B for 2005/179 and 2003/651 than for the other strains, suggesting a higher degree

**Figure 5 | Activity of fluoroquinolones alone or combined with caspofungin on *S. aureus* biofilms *in vivo*.** (**a–f**) Biofilms were formed by the clinical isolate 2011S027 (**a,d**), by the bioluminescent strain Xen36 (**b,e**) or by the clinical isolate 2003/651 (**c,f**) in the mouse subcutaneous biofilm model. Animals were treated for 7 days with caspofungin ($4\,mg\,kg^{-1}$ of body weight) once daily, fluoroquinolones ($40\,mg\,kg^{-1}$ of body weight) twice daily, or with a fluoroquinolone and caspofungin (each injected separately and according to its own schedule). CAS: caspofungin; MXF: moxifloxacin; DFX: delafloxacin. Control animals (CTRL) were injected with normal saline (0.9% NaCl) twice daily. (**a–c**) Experiments with moxifloxacin; (**d–f**) experiments with delafloxacin. (**g**) Dose–response for the activity of delafloxacin alone or combined with caspofungin on biofilms from the clinical isolate 2011S027 in the mouse subcutaneous biofilm model. Animals were treated for 7 days with caspofungin (CAS; $4\,mg\,kg^{-1}$ of body weight) once daily, delafloxacin (DFX; 10, 20 or $40\,mg\,kg^{-1}$ of body weight) twice daily, or with delafloxacin at each of these doses combined with caspofungin $4\,mg\,kg^{-1}$ once daily. Data are presented as the number of CFU (in $\log_{10}$ units) recovered from each individual catheter, with the mean value (and s.e.m. in **g**) shown in green or red colour for all catheters (five catheters per animal; three or four animals per group; one experiment with clinical isolates and three experiments with strain Xen36). Statistical analysis (one-way ANOVA; Tukey *post-hoc* test): groups with different letters are significantly different from one another ($P < 0.05$).

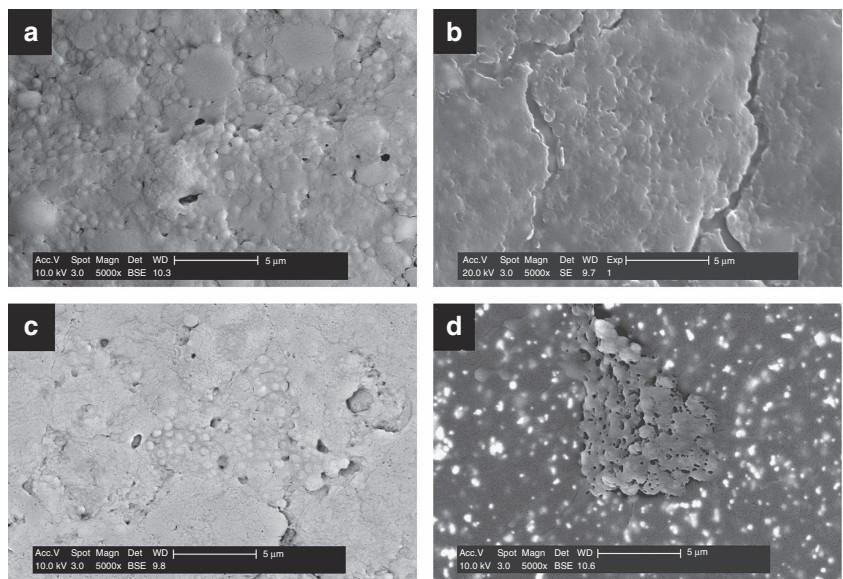

**Figure 6 | Scanning electron microscopy images of *S. aureus* 2011S027 biofilms developed *in vivo* inside polyurethane catheters pieces.** Fragments were retrieved from animals treated with sterile saline (**a**; control group), 4 mg kg$^{-1}$ caspofungin once daily (**b**), 40 mg kg$^{-1}$ moxifloxacin twice daily (**c**) or their combination (**d**). Scale bars, 5 μm.

of PNAG polymerization in the corresponding biofilms. In caspofungin-treated biofilms, the concentration of monomers was reduced to 50–60% in all strains except 2003/651, which remained unaffected. This strongly suggests that caspofungin impairs *N*-acetylglucosamine polymerization in biofilms formed from strains that are susceptible to its effects.

Biofilm matrix also contains other constituents like extracellular DNA and proteins. Their concentration was therefore also evaluated in biofilms that were incubated during 48 h with 40 mg l$^{-1}$ caspofungin versus controls, but no effect was observed, indicating that the action of caspofungin was not unspecific (Supplementary Fig. 4).

**Inhibition of *N*-acetylglucosamine transferase by caspofungin**. Poly-β(1-6)-*N*-acetylglucosamine is synthesized by enzymes encoded by the *ica* operon comprising four genes (*icaA*, *icaD icaB* and *icaC*). Among them, *icaA* encodes a membrane-located *N*-acetylglucosamine transferase that catalyzes the addition of new *N*-acetylglucosamine monomers to the growing polymer (Fig. 9a)[13]. The mode of action of caspofungin as antifungal agent is to prevent the incorporation of uridine diphosphate (UDP)-glucose into β-1-3-glucan by inhibiting the fungal β-1-3-glucan synthase[29]. A BLAST and clustalW analysis revealed conserved regions between the sequence of *S. aureus icaA* and that of the genes encoding β-1-3-glucan synthase from diverse fungal species (Supplementary Fig. 5). Interestingly, these regions correspond to conserved amino acids described as catalytic residues (Asp[134]; Asp[227]; Arg[276]) in IcaA[13]. On the basis of the data presented in Fig. 8, which strongly suggest an effect of caspofungin on *N*-acetylglucosamine incorporation in growing PNAG polymers, we investigated whether caspofungin could inhibit IcaA activity. We first compared the enzymatic activity in protein extracts prepared from strain ATCC33591 or its Δ*icaA* mutant (Fig. 9b) and found that the enzymatic activity of the extract from the wild-type strain was 5.62 IU mg$^{-1}$ against only 0.02 IU mg$^{-1}$ for the mutant extract. Notably, the enzymatic activity of the wild-type extract was markedly inhibited in the presence of 40 mg l$^{-1}$ caspofungin (residual activity: 0.10 IU mg$^{-1}$). This led us to conclude that caspofungin can inhibit bacterial *N*-acetylglucosamine transferase activity.

We therefore examined the effect of increasing concentrations of caspofungin on UDP release by extracts from ATCC33591 and 2003/651. These two strains were selected because they express *icaA*, respectively, to the lowest and the highest level among the studied strains (see Supplementary Table 2; this table also shows the expression levels of the other genes of the *ica* operon). In parallel, we also sequenced *icaA* in ATCC33591 and 2003/651, looking for potential mutations that could explain differences in caspofungin inhibition towards IcaA activity from these two strains. As illustrated in Supplementary Fig. 6, only six silent mutations were found in 2003/651, which did not alter the amino acid sequence but could contribute to explain the high expression level of *icaA* in this strain by an exchange in the used codon. It has indeed been observed in several species that gene expression levels tend to correlate with the codon usage[41] and that rare codons increase 4 to 20-fold gene expression levels[42,43] possibly by making easier ribosome trafficking throughout the coding sequence[44].

As illustrated in Fig. 9c, a clear concentration-effect for the inhibition of IcaA activity by caspofungin was obtained for both strains, with, however, a major difference in EC$_{50}$ values (1.6 versus 9 mg l$^{-1}$, respectively). Considering then the eight strains under investigation, we looked for a possible correlation between (a) the concentrations of fluoroquinolones needed to achieve a specific reduction in viability within biofilms when combined with caspofungin and (b) the level of *icaA* expression. We selected as target effect a reduction in viability of 25% for moxifloxacin and of 50% for delafloxacin, because the latter was more active than the former when used alone. A highly significant correlation (Pearson's correlation coefficient > 0.84) between these two parameters was observed (Fig. 9d,e).

**Discussion**
Because of the difficulties of eradicating bacterial infections with current antibiotic treatments once biofilms are formed, discovering innovative strategies that specifically enhance antibiotic efficacy in this setting contributes to fill a therapeutic gap and therefore answers a clear unmet medical need[24,45,46]. Our work contributes to this effort by demonstrating the adjuvant properties of caspofungin towards fluoroquinolone activity

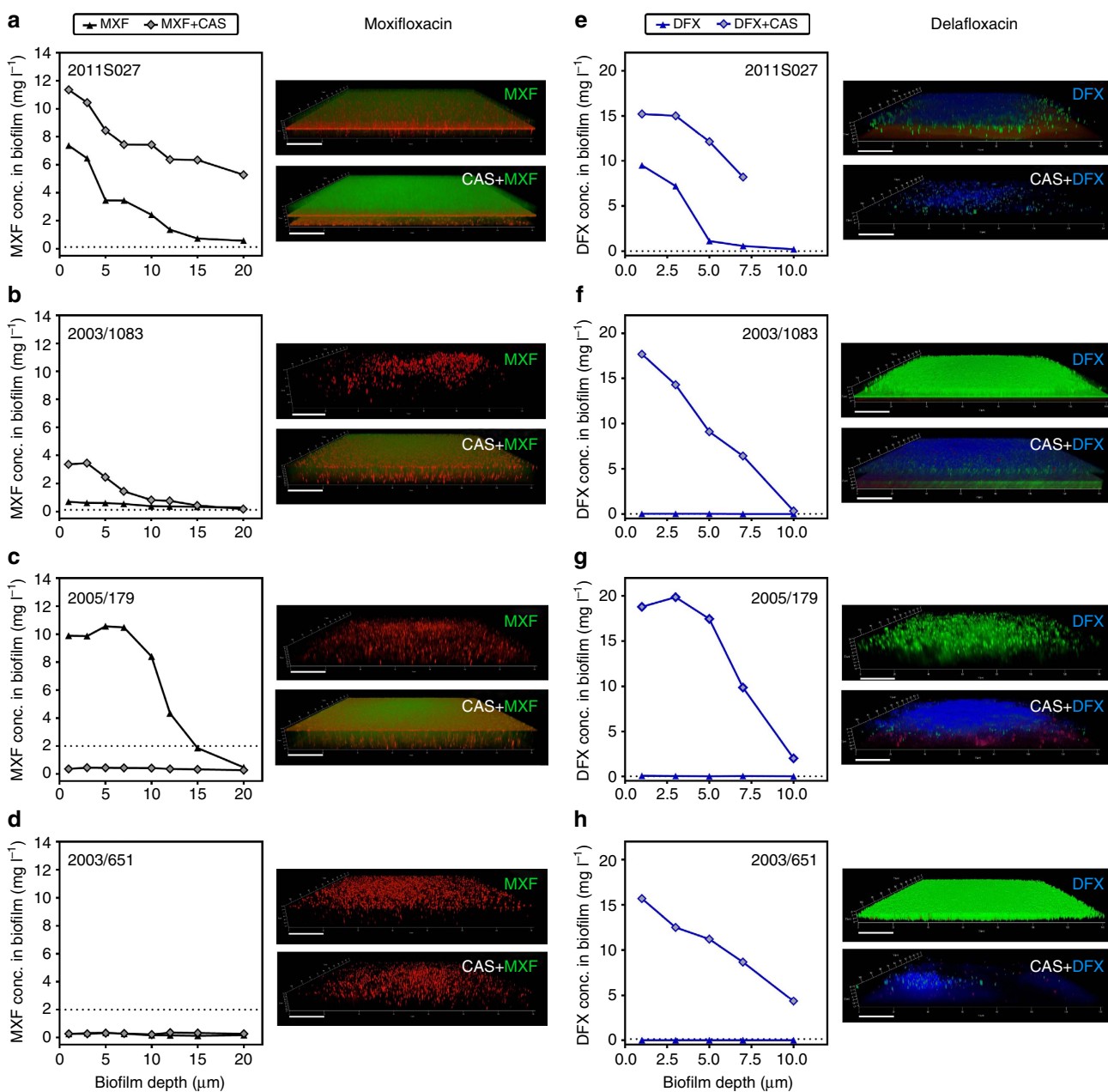

**Figure 7 | Effect of caspofungin on fluoroquinolone penetration within biofilms.** (**a–d**) Moxifloxacin (MXF); (**e–h**) delafloxacin (DFX). The graphs compare the concentration of fluoroquinolone in biofilms incubated with 20 mg l$^{-1}$ of fluoroquinolone alone (MXF or DFX) or combined with 40 mg l$^{-1}$ caspofungin (MXF + CAS or DFX + CAS). The horizontal dotted line corresponds to the fluoroquinolone MIC for the corresponding strain. The tridimensional images were obtained using confocal laser scanning microscopy for the corresponding biofilms stained either with 0.5 mM of 5-cyano-2,3-ditolyl tetrazolium chloride (red signal: MXF experiment) or with LIVE/DEAD (green signal: living bacteria; red: dead bacteria). The moxifloxacin fluorescence signal appears as green (preventing us from using LIVE/DEAD staining of the corresponding biofilm) and the delafloxacin signal, in blue. Scale bars, 20 μm.

against *S. aureus* biofilms. This synergy is observed not only in two *in vitro* models (biofilms growing in 96-well plates or on catheters), but also *in vivo,* using a series of clinical isolates that were previously demonstrated as poorly susceptible to these and to other antibiotics when growing as biofilms *in vitro*[30].

Caspofungin is described as an antifungal agent with no intrinsic antibacterial activity. We present here three pieces of convergent experimental evidence that caspofungin increases the activity of fluoroquinolone by destructuring *S. aureus* biofilm matrix through an inhibition of the bacterial *N*-acetylglucosamine transferase (IcaA). First, caspofungin inhibits *in vitro* the

enzymatic activity of IcaA in the range of concentrations at which it also increases fluoroquinolone activity on biofilms. Second, caspofungin decreases the concentration and the degree of polymerization of poly-β(1-6)-*N*-acetylglucosamine in biofilms. Third, while the degree of synergy between caspofungin and fluoroquinolones markedly differs among the two antibiotics tested as well as among strains, it correlates with the respective level of expression of *icaA* in these strains. Although unexpected, inhibition of bacterial IcaA by caspofungin can be rationalized by the fact that it is a homologue of β-1-3-glucan synthase, the fungal target of caspofungin. While caspofungin proved effective

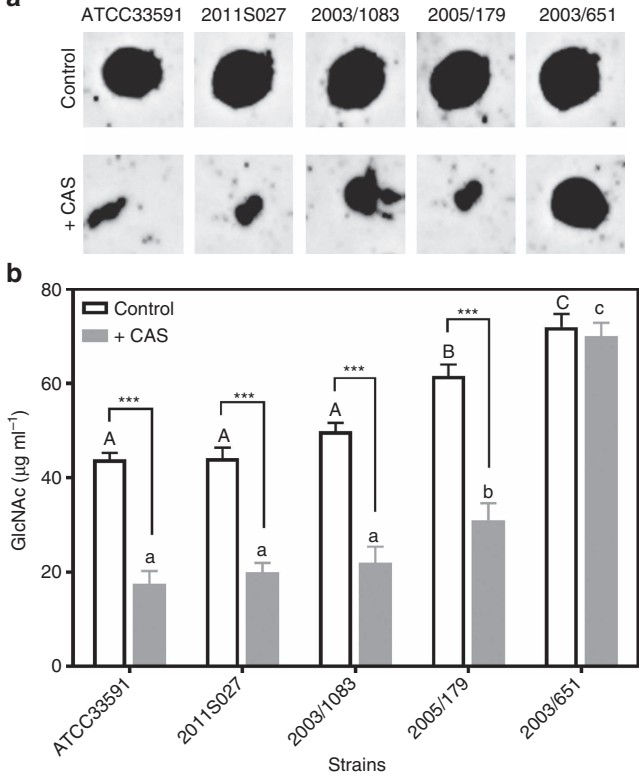

**Figure 8 | Effect of caspofungin on poly-β(1-6)-N-acetylglucosamine in biofilms.** Biofilms were incubated during 24 h in the absence (control) or in the presence (+CAS) of 40 mg l$^{-1}$ caspofungin. (**a**) Immunoblot analysis of PNAG purified from biofilms. (**b**) Determination of N-acetyl-glucosamine (GlcNAc) concentration after treatment of purified PNAG during 1 h with 0.1% Dispersin B. The released GlcNAc monomers were detected by fluorescence (λexc 545 nm; λem 604 nm). All data are mean ± s.d. of triplicates. Statistical analysis (two-way ANOVA; Tukey post-hoc test): groups with different letters (caps: control; small letters: +CAS) are significantly different from one another ($P < 0.05$); *** ($P < 0.001$): comparison of control to CAS for each individual strain.

to inhibit poly-β(1-6)-N-acetylglucosamine polymerization in bacterial membrane extracts and to disperse it within biofilms, it only exerted modest effects on the biomass or on the three-dimensional structure of biofilms in the absence of moxifloxacin. This suggests that bacterial killing (by an antibiotic) is required to observe an effective disruption of the biofilm. This hypothesis is coherent with the observation that bacterial death within biofilms is associated with a facilitation of biofilm dispersal, related to the creation of voids within the matrix[47,48]. The main role of caspofungin, therefore, would be to increase the ability of fluoroquinolones to penetrate more deeply into the biofilm by decreasing the amount and/or the degree of polymerization of N-acetyl-glucosamine polymer present in the matrix network. We previously demonstrated that the bacterial killing exerted by delafloxacin towards bacteria present in biofilms formed by the same strains is strictly dependent on its capacity to penetrate the matrix[30]. We complement this observation here by showing that its killing activity is much lower against those strains that express icaA to high levels, in close correlation with its decreased penetration in the biofilm. We also extend this observation to another fluoroquinolone, moxifloxacin. This is coherent with the fact that, beside their contribution in bacterial adhesion and aggregation, exopolysaccharides are also critical for the maintenance of the biofilm architecture and viscoelastic

properties[49,50], playing, therefore, a key role in limiting antibiotic penetration. Supporting this specific role for exopolysaccharides, planktonic cultures of S. aureus that spontaneously form aggregates because of a production of these polymers have been also shown to be refractory to antibiotic activity, but to regain susceptibility on disruption of the aggregates by sonication[51]. Thus, caspofungin can be considered as a dispersal agent capable of improving antibiotic activity against biofilms in a similar way as enzymes such as dispersin B, proteinase K or DNase I[22,52,53]. However, we show here that its association with a fluoroquinolone antibiotic is essential for maximal efficacy. The key advantage of combining a dispersal agent and a bactericidal antibiotic is, indeed, that the latter also acts on planktonic bacteria, which may avoid the spreading of bacterial clumps released from the matrix on dispersal and the subsequent reestablishment of a biofilm elsewhere.

Our work has two main limitations. First, synergy was mainly demonstrated for fluoroquinolones and was not observed or only in a limited fashion for two other well established anti-staphylococcal antibiotics. For vancomycin and daptomycin, this may partly be due to their large molecular mass (1,449 and 1,620 g mol$^{-1}$, respectively) versus fluoroquinolones (~400 g mol$^{-1}$), that may hamper their ability to diffuse into the biofilm[30] even if disrupted by caspofungin. For linezolid, for which the molecular mass (337 g mol$^{-1}$) is lower than that of fluoroquinolones, this could be due to its bacteriostatic effect against S. aureus, thus limiting its overall activity. Beside antibiotic properties, we cannot exclude that other matrix properties can contribute to prevent antibiotic action. Yet, we did not find any relationship between antibiotic loss of activity and the biofilm content in other major constituents like proteins or extracellular DNA. Likewise, we did not observe significant differences among strains in the expression levels of icaC, which encodes an O-succinyltransferase that is thought to influence biofilm accumulation[15]. A second limitation of our study is that we only examined biofilms formed from S. aureus, whereas those formed by related species such as Staphylococcus epidermidis are also clinically relevant. Interestingly, this organism also expresses the ica locus and produces a poly-β(1-6)-N-acetylglucosamine-rich matrix[54,55].

In spite of these limitations, and although remaining focused on in vitro and animal demonstrations, our finding may have important clinical implications. First, the synergistic effects observed in vivo were obtained while using caspofungin and fluoroquinolones at doses that are clinically relevant, suggesting they could be also observed in humans. Second, S. aureus strains expressing the ica locus are highly prevalent in biofilm-related infections. A recent study showed that 85.6% of strains collected from human or bovine infections expressed ica genes, among which 95.4% were biofilm producers[18]. Likewise, in a collection of strains causing catheter-related urinary infections, 88.6% were biofilm-producers and all of them expressed the ica locus[16]. In this context, an Argentinian study showed that 35% of strains collected in nasal swabs from hospitalized patients or from staff are ica-positive and that all were slime producers[56]. Conversely, ica-negative strains producing slime have been rarely described in S. aureus infection[57], underlining the interest of detecting icaA expression in clinical isolates prior using inhibitors of IcaA as adjuvant therapy. Third, we document the promising activity of delafloxacin, a fluoroquinolone in clinical development, against staphylococcal biofilms, especially when combined with caspofungin against strains expressing icaA to high levels. Fourth, and perhaps more importantly, no usable pharmacological inhibitor of IcaA has been described so far, and we have identified here an agent that is already approved for clinical use in

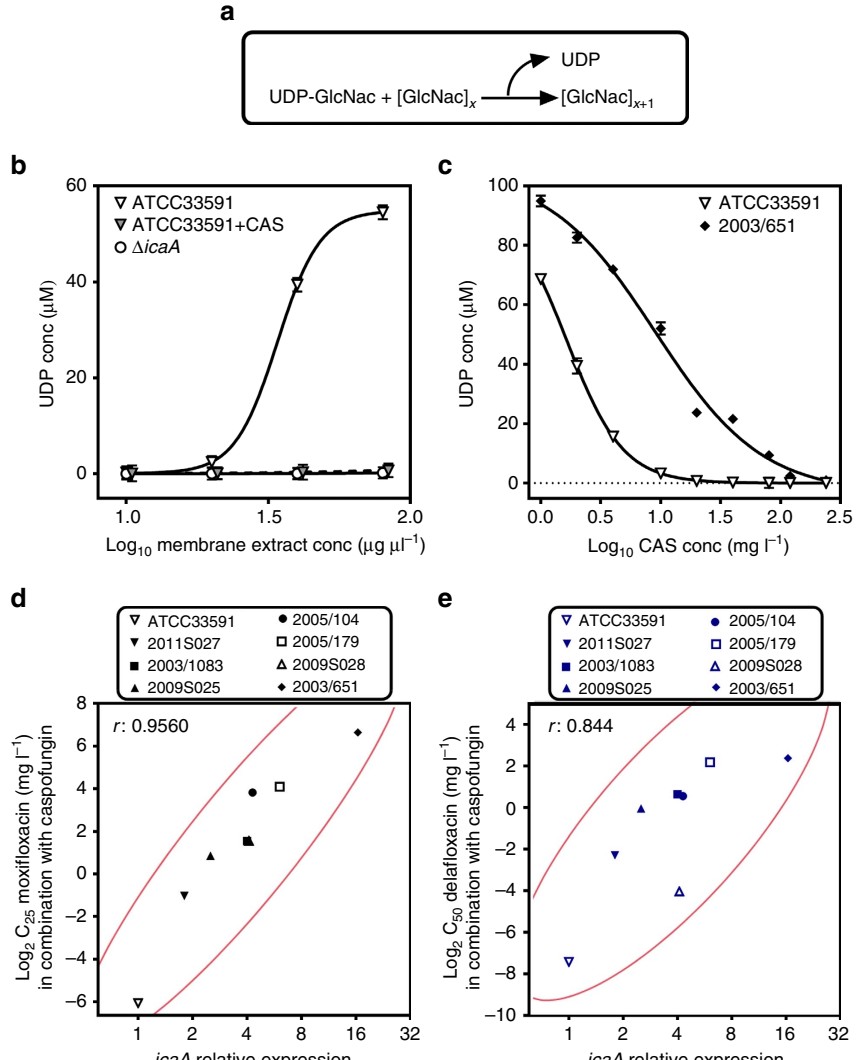

**Figure 9 | Ica A (*N*-acetylglucosamine transferase) activity and its inhibition by caspofungin.** (**a**) Reaction catalyzed by IcaA. (**b**) Activity of IcaA in membrane protein extracts from strain ATCC33591 and its $\Delta icaA$ mutant; increasing concentrations of extracts (1.25–80 $\mu g \mu l^{-1}$) were incubated with 40 mM UDP-GlcNAc and 400 mM *N*-acetylglucosamine without (ATCC33591 and $\Delta icaA$) or with 40 mg $l^{-1}$ caspofungin (ATCC33591 + CAS) during 120 min; activity was evaluated by the amount of UDP liberated in the reaction medium; all data are means ± s.d. of triplicates. (**c**) Inhibition of IcaA activity in membrane protein extracts from strain ATCC33591 or the clinical isolate 2003/651 (both at 80 $\mu g \mu l^{-1}$) exposed to increasing concentrations of caspofungin (CAS) during 120 min; all data are means ± s.d. of triplicates; (**d,e**): correlation between the *icaA* relative expression and the relative potency of the fluoroquinolone–caspofungin combinations against biofilms. Each symbol corresponds to a specific strain. Fluoroquinolone potency is expressed as the concentration needed to reduce of 25% ($C_{25}$; for moxifloxacin (**d**)) or of 50% ($C_{50}$; for delafloxacin (**e**)) the bacterial viability within biofilms exposed to the fluoroquinolone combined with 40 mg $l^{-1}$ caspofungin (data from Fig. 2). The graph shows the actual data (symbols; bivariate fit) surrounded by the bivariate normal ellipse for 95% confidence interval ($r$ is the Pearson's correlation coefficient).

fungal infections. Although caspofungin is a low-affinity inhibitor of IcaA, our results warrant further studies to validate such inhibitors as potential adjuvants for antibiofilm antibiotherapy.

We fully realize that administering caspofungin to patients who are not infected by fungi could affect the fungal flora. We hope that our work will stimulate research for development of more potent IcaA inhibitors acting specifically on the staphylococcal enzyme and devoid of antifungal activity.

## Methods

**Materials.** Microbiological standards or solutions for injection were obtained from the following sources: moxifloxacin HCl (powder potency: 90.9%, from Bayer HealthCare; Leverkusen, Germany; a solution for injection was prepared in NaCl 0.9%); B.5.delafloxacin (powder potency: 95.7%) and its intravenous formulation, from Melinta Therapeutics (New Haven, CT, USA); caspofungin diacetate (powder potency: 90.1%), from Sigma-Aldrich (St Louis, MO, USA) and Cancidas, from

MSD (Brussels, Belgium). The other antibiotics were used as powder or solution for injection approved for human use in Belgium and complying with the provisions of the European Pharmacopoeia (vancomycin as Vancomycine Mylan, Mylan Inc, Canonsburg, PA, USA; linezolid as Zyvoxid, Pfizer Inc. (New York, NY, USA); daptomycin as Cubicin, Novartis (Horsham, UK)). Media for bacterial culture were from Becton Dickinson Company (Franklin Lakes, NJ, USA).

**Bacterial strains.** The *S. aureus* ATCC33591 (methicillin-resistant) strain was used as reference strain. Seven clinical strains isolated from various human anatomical sites but all belonging to the pandemic clonal complexes CC5 or CC8 of *S. aureus* were selected from the collection of the Belgian Reference Centre for *S. aureus* (Hôpital Erasme, Université libre de Bruxelles, Brussels) (Table 1). The bioluminescent strain Xen36 (Caliper Life Sciences, Hopkinton, MA, USA) was originally derived from *S. aureus* ATCC49525 and expresses a stable copy of a modified *Photorhabdus luminescens luxABCDE* operon[58,59]. MICs were determined by microdilution assay according to the recommendations of the Clinical & Laboratory Standards Institute[60] (Table 1).

**In vitro biofilm model.** Mature biofilms were obtained by growing bacterial strains for 24 h in 96-well tissue culture plates (VWR (Radnor, PA, USA); European cat. number 734-2327) in Trypticase Soy Broth supplemented with 2% NaCl and 1% glucose, with a starting inoculum adjusted to an optical density at 620 nm ($OD_{620nm}$) of 0.005 in a volume of 200 μl, as previously described[32]. Biofilms were then exposed for 48 h to antibiotics (concentration range: 0.125–20 mg l$^{-1}$) alone or in combination with caspofungin (40 mg l$^{-1}$). Bacterial viability in the biofilm was measured using the redox indicator resazurin, as previously described[30,32], or by CFU counting. To this effect, biofilms were washed twice with phosphate buffer saline (PBS), sonicated (Branson 5510 Ultrasonics bath) for 10 min in 1 ml PBS and diluted aliquots were plated on tryptic soy agar (TSA) plates to allow CFU counting after overnight incubation. Biofilm biomass was determined using crystal violet staining, following a published procedure[30,32]. In brief, at the end of the incubation period, the medium was removed and wells were washed with PBS, fixed at 60 °C for 1 h and stained by a 2.3% crystal violet solution prepared in ethanol 20% (Sigma-Aldrich). After elimination of the dye in excess under running water, crystal violet fixed to the biofilm was resolubilized by addition of 33% glacial acetic acid and incubation at room temperature for 1 h. Absorbance was read at 570 nm.

**Biofilm grown on catheters in vitro.** Biofilms were studied inside triple-lumen polyurethane central venous catheters (Certofix duo/trio; B. Braun Melsungen AG, Melsungen, Germany) as described earlier[61]. Briefly, 1 cm long catheters were incubated overnight in 100% fetal bovine serum (Sigma-Aldrich). Catheters were incubated 90 min at 37 °C in trypticase soy broth supplemented with 2% NaCl and 1% glucose with bacterial strains at an initial density of 0.005 to allow adhesion, then transferred to new medium and incubated during 24 h. Biofilms were then exposed to 10 mg l$^{-1}$ fluoroquinolone, 40 mg l$^{-1}$ caspofungin or their combination for 48 h. Catheter pieces were washed with PBS, sonicated and diluted before plating on TSA and CFU counting, as described in the previous paragraph.

**Murine subcutaneous biofilm model.** Female pathogen-free 20 g 8-week old BALB/c mice (Janvier Labs, Saint Berthevin, France) were kept individually in ventilated cages and provided with food and water ad libitum. All animal experiments were performed in accordance with the regulations and approval of the Ethical Committee of KULeuven (project number P125/2011). Animals were immunosuppressed by adding 0.4 mg l$^{-1}$ dexamethasone (Organon Laboratories Limited, Cambridge, UK) in their drinking water 24 h before the catheter implant and during the whole experiment. Biofilms were studied using clinical isolates (2011S027 or 2003/651) or the bioluminescent strain (Xen36). Serum-coated catheters were incubated with the bacteria during the period of adhesion (90 min at 37 °C) as described above. Afterwards, catheters were washed twice with PBS and subsequently implanted subcutaneously in the back of mice as described hereunder. First, general anaesthesia was achieved by intraperitoneal injection of a mixture of 45 mg kg$^{-1}$ ketamine (Ketamine1000; Pfizer, Puurs, Belgium) and 0.6 mg kg$^{-1}$ medetomidine (Domitor; Pfizer) and local anaesthesia by application of a 2% xylocaine (AstraZeneca BV, Zoetermeer, Netherlands) on the skin. The lower back of the mice was then shaved and disinfected with 0.5% chlorhexidine in 70% alcohol. A 10 mm incision was made longitudinally and five catheter fragments were implanted per mouse. Biofilms were allowed to mature in vivo for 24 h before treatment. Fluoroquinolones and caspofungin were administered intraperitoneally. Whereas caspofungin was administered once daily (4 mg kg$^{-1}$ of body weight), fluoroquinolones (up to 40 mg kg$^{-1}$ of body weight) were injected twice daily for 7 days. A control group of animals was injected twice daily with saline only. After 7 days of treatment, animals were killed by cervical dislocation and catheters were removed, washed twice with PBS (to remove non-device associated bacteria) and sonicated. The number of viable bacteria recovered from the biofilms was then quantified by CFU counting after plating and overnight growth at 37 °C.

**In vivo bioluminescent imaging.** Biofilms made with the bioluminescent strain Xen36 were prepared and animal treated exactly as described above. Mice were imaged every day using an In Vivo Imaging System (IVIS Spectrum, Perkin-Elmer, Waltham, MA, USA). During the imaging, mice were anaesthetized using a gas mixture of isoflurane in oxygen (1.5–2%) and placed by groups of four animals in the apparatus. Frames were acquired with a field of view of 23 cm. Consecutive scans with acquisition time of 5 min (binning 2) were acquired until maximal signal intensity was reached. The signal was quantified by using Living Image software (version 4.0, Perkin-Elmer) and reported as photon flux per second (p s$^{-1}$) for a rectangular region of interest placed over each five catheters)[62].

**Scanning electron microscopy.** Mounted samples were sputter-coated with Au–Pd and viewed using a scanning electron microscope operated at standard high vacuum settings at a 10-mm working distance and 10-keV accelerating voltage (FEI XL30-FEG microscope, Philips Nederland B.V., Eindhoven, the Netherlands).

**Confocal laser scanning microscopy for visualization of biofilms.** Biofilm samples were imaged using a Cell Observer s.d. confocal fluorescent microscope (Carl Zeiss AG, Oberkochen, Germany) using spinning disc technology (Yokogawa Electric Corporation, Tokyo, Japan) and controlled by the AxioVision software

(AxioVs40 V 4.8.2.0; Zeiss). Optimal confocal settings (camera exposure time, CSU disk speed) were determined in preliminary experiments. Image stacks of each sample were acquired at a resolution of 700 × 500 pixels and recorded using Z-Stack module for acquisition of image series from different focus planes and used to construct three-dimensional images with AxioVision software.

**Fluoroquinolone penetration within biofilms.** Twenty-four hour biofilms were grown on cover slips and incubated for 1 h with 20 mg l$^{-1}$ fluoroquinolone alone or in combination with 40 mg l$^{-1}$ caspofungin. Biofilms were washed twice with 1 ml PBS and stained for 30 min in the dark with LIVE/DEAD bacterial viability kit (L-7007; Thermo Fisher Scientific, Waltham, MA, USA) or with 0.5 mM 5-cyano-2,3-ditolyl tetrazolium chloride (CTC) (RedoxSensor vitality kit; Invitrogen, Carlsbad, CA, USA), as described previously[30]. CTC is a colourless, non-fluorescent and membrane permeable compound, which is readily reduced via electron transport activity to fluorescent, insoluble CTC-formazan that accumulates inside bacteria[63]. Stained biofilms were then washed with 1 ml PBS buffer. Excitation/emission wavelengths were set as follows: 415 nm/500–550 nm for moxifloxacin; 395 nm/450 nm for delafloxacin, 488 nm/570–620 nm for CTC-formazan signal, 488 nm/500–550 nm for Syto 9 and 561 nm/570–620 nm for propidium iodide (LIVE/DEAD staining). Fluoroquinolone concentrations within biofilms were then calculated using calibration curves built using fluoroquinolone solutions (concentrations ranging from 5 to 50 mg l$^{-1}$) examined in the microscope using the same settings as for samples[30].

**PNAG purification and immunoblot analysis.** PNAG was extracted as previously described[64]. Briefly, biofilms were resuspended in PBS, centrifuged and resuspended in 0.5 M EDTA, and incubated at 100 °C for 5 min and at 85 °C for 30 min. After a new centrifugation, the supernatant was first dialyzed against deionized water for 18 h and then against a buffer (50 mM Tris–HCl pH 8; 20 mM $MgCl_2$) for 18 h (cut-off of dialysis membrane: 2 kDa). The crude polysaccharide preparation was treated with 100 μg ml$^{-1}$ α-amylase, 500 μg ml$^{-1}$ lysozyme, 250 μg ml$^{-1}$ DNase I and 100 μg ml$^{-1}$ RNase A at 37 °C for 2 h, then by 2 mg ml$^{-1}$ proteinase K for 16 h at 55 °C in the presence of 1 mM $CaCl_2$ and 0.5% sodium dodecyl sulfate. The samples were then incubated at 85 °C for 1 h to inactivate proteinase K and dialyzed against deionized water for 18 h. Polysaccharide preparations were then lyophilized and dissolved in 200 μl PBS. A 20 μl aliquot was spotted onto a PVDF membrane, which was air-dried and blocked with 0.5% milk in TBS buffer (150 mM NaCl and 10 mM Tris–HCl (pH 7.4)) overnight at 4 °C. The membrane was then incubated overnight at 4 °C with PNAG antiserum (1:4,000)[65] (kindly provided Dr Gerald B. Pier; Brigham and Women's Hospital, Boston, MA, USA), washed and probed with 1:5,000 goat anti-rabbit HRP for 2 h. Spots were visualized with the SuperSignal West Pico Chemiluminescent Substrate kit (Thermo Fisher Scientific) and analyzed using the FUSION-CAP Software (Analis, Belgium).

**GlcNAc determination by Morgan–Elson assay with fluorimetric detection.** Purified PNAG (125 μl) was incubated with 0.1% dispersin B (Symbiose, Belgium) for 1 h at 37 °C to cleave them in N-acetylglucosamine (GlcNAc) monomers (note that the dialysis steps performed during the purification procedure allowed to eliminate the pre-formed monomeric forms). These were quantified by the Morgan–Elson reaction[66]. The sample was added by 25 μl of tetraborate reagent ($K_2B_4O_7 \cdot 4H_2O$ 0.8 M; 3 min at 100 °C), then by 0.75 ml of p-dimethylaminobenzaldehyde reagent (prepared as described[67]; 10 min incubation at 37 °C). The released reducing terminal GlcNAc in the supernatant were quantified in the supernate (recovered after 30 s centrifugation at 8,000g) by fluorimetry (λexc: 545 nm; λem: 604 nm)[68] based on a calibration curve constructed using GlcNAc standards (Sigma-Aldrich).

**Assay of proteins and DNA in biofilms.** Biofilms were grown in 6-wells plates during 24 h after which they were incubated during 48 h in the presence of 40 mg l$^{-1}$ caspofungin or in control conditions. Extracellular DNA (eDNA) was quantified as previously described[69]. In brief, biofilms were washed twice and chilled at 4 °C for 1 h, after which 500 μl of 0.5 M EDTA was added to each well. After centrifugation (5 min, 12,000g), supernatants were discarded, and biofilms were resuspended in eDNA extraction solution (50 mM Tris · HCl, 10 mM EDTA, 500 mM NaCl, pH 8) and transferred into chilled tubes. After centrifugation (5 min; 4 °C; 18,000g), 100 μl of supernatant was transferred to a tube containing 300 μl of TE buffer (10 mM Tris · HCl, 1 mM EDTA, pH 8), and extracted once with an equal volume of phenol/chloroform/isoamyl alcohol (25:24:1) and once with chloroform/isoamyl alcohol (24:1). The aqueous phase of each sample was then mixed with three volumes of ice-cold 100% (vol/vol) ethanol and 1/10 volume of 3 M sodium acetate (pH 5.2) and stored at − 20 °C. The next day, the ethanol-precipitated DNA was collected by centrifugation (20 min; 4 °C; 18,000 g), washed with ice-cold 70% (vol/vol) ethanol, air-dried and dissolved in 20 μl TE buffer. DNA concentration was then determined with NanoDrop Ultraviolet–vis spectrophotometer (Thermo Fisher Scientific). Total proteins were quantified following a described procedure[70]. Biofilm cells were pelleted by centrifugation at 5,500g for 10 min. Proteins present in the supernatant were precipitated by trichloroacetic acid (TCA). Biofilm cells were gently resuspended in 5 ml PBS

(pH 10) containing complete Protease Inhibitor Cocktail (Roche Life Science, Penzberg, Germany) (according to manufacturer's instruction). Cells were incubated at 4 °C with gentle rotation for 1 h and debris pelleted by centrifugation at 5,500g for 10 min. Proteins for biofilms supernatants and biofilms cells were then assayed using Bradford method.

**Determination of the N-acetylglucosamine transferase activity in vitro.** Protein extracts from membranes of ATCC33591 and of its ΔicaA mutant (a kind gift from Prof. Friedrich Götz, Universität Tübingen, Germany) were prepared as follows. Fifty ml of overnight cultures were collected by centrifugation, and cell pellets were resuspended in 500 μl of buffer A (50 mM Tris–HCl, pH 7.5, 10 mM MgCl₂, 4 mM dithiothreitol). Cells were disrupted by 3 × 1 min vortexing in Corex tubes with glass beads (Sigma-Aldrich) (diameter of 0.3–0.60 mm; two times the weight of the cell pellet). DNase I (20 μg ml⁻¹) was added before breaking the cells. Unbroken cells and glass beads were sedimented (10 min, 2,000g), and the supernatant was saved. The procedure was repeated twice and all supernatants were combined. Membranes were sedimented from the crude extract by centrifugation (40 min, 20,000g), resuspended in buffer A, extracted with 2% (w/v) Triton X-100 (in buffer A) for 2 h with gentle shaking, sedimented again, washed once with buffer A and resuspended in 1 ml of buffer A. Protein concentrations were determined by the Lowry's method.

To determine N-acetylglucosamine transferase activity, increasing concentrations (1.25–80 μg μl⁻¹) of membrane extracts were incubated for 2 h at 37 °C in 250 mM MES-NaOH buffer pH 6.25 containing 40 mM UDP-GlcNAc, 20 mM MnCl₂, 400 mM N-acetylglucosamine, 1% (w/v) Triton X-100, in a total volume of 50 μl. The reaction was stopped by addition of 50 μl of Milli-Q water and boiling for 2 min. The mixture was then centrifuged at 20,000g for 5 min to remove denatured proteins and then supernatant was collected. The N-acetylglucosamine transferase activity was evaluated by measuring the amount of UDP liberated in the reaction using the Transcreener UDP² FP Kit (BellBrook Labs, Madison, WI, USA), according to the manufacturer's recommendations. Fluorescence polarization was measured using a Perkin-Elmer LS55 fluorimeter (Perkin-Elmer). UDP concentration was then calculated based on a titration curve established as described in Supplementary Fig. 7.

**RNA isolation from biofilms and quantitative real-time PCR.** RNAs were isolated from 24 h-old biofilms. Biofilms formed in 6-well polystyrene plates were washed thrice with sterile distilled water. Bacterial cells were detached by rapid scraping and resuspended in cold sterile distilled water. Suspensions were immediately incubated with 1 ml of RNA protect (Qiagen GmbH, Hilden, Germany), vortexed for 5 s and incubated for 5 min at room temperature, and pelleted by centrifugation at 10,000g for 10 min. Cell pellets were resuspended in 100 μl of 4 °C sterile RNase-free distilled water (Qiagen). Total RNA was isolated using RNeasy mini kit (Qiagen). The RNA quality and quantity was checked by agarose gel electrophoresis and by measuring the absorbance at 260 and 280 nm using a NanoDrop™ spectrophotometer. Purified RNA was immediately converted to cDNA using transcription first strand cDNA synthesis kit (Roche Applied Science) with random hexamer primers according to the manufacturer's instructions. Quantitative PCR reactions were performed in triplicates in 96-well plates using 2 μl of cDNA, 10 μl of SYBR Green Master Mix, 0.5 μl of 100 μM of each primer, and 7 μl of sterile RNase-free water. The following primers were used: icaA gene forward (5′-CGAGAAAAAGAATATGGCTG-3′) and reverse (5′-ACCATGTTGC GTAACCACCT-3′); 16s rRNA gene forward (5′-CGAAGGCGACTTTCTGG TCT-3′) reverse (5′-TACTCCCCAGGCGGAGTGCT-3′). The reaction was started with an initial denaturation at 95 °C for 5 min, followed by 40 amplification cycles of 95 °C for 20 s, 60 °C for 20 s and 72 °C for 20 s. The X-fold change of transcription level was calculated using a relative standard curve method as described previously[71].

**Sequencing icaA gene.** cDNA from the ATCC33591 strain and from the 2003/651 isolate were amplified via PCR using Phusion High-Fidelity DNA Polymerase following manufacturer's protocol (Thermo Fischer Scientific). The primers used for DNA amplification were icaAfwd:5′-GTTATCAATAATCTTATC CTT-3′ and icaArev: 5′-AGTTTCAAATATATCTAAAAT-3′. The PCR product (1611pb) was sequenced at Beckman Coulter Genomics facilities (Beckman Coulter Genomics, Takeley, Essex, UK) following Sanger protocol. The primers used for DNA sequencing were icaAfwd; icaArev and seqicaAREV: 5′-CCAATGTTTCT GGAACCAACA-3′.

**Data analyses and statistical analyses.** Curve-fitting analyses of concentration-effect relationships were made with GraphPad Prism version 4.03 or version 7.01 (GraphPad Software, San Diego, CA, USA) and correlations with JMP Pro version 12.1.0 (SAS Institute, Marlow, Buckinghamshire, UK). Statistical analyses were made with GraphPad Instat version 3.06 (GraphPad Software) or GraphPad version 7.01.

**Data availability.** The authors declare that the data supporting the findings of this study are available within the article and its Supplementary Information files.

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

## Acknowledgements

W.S. was a postdoctoral fellow of the programme Prospective Research for Brussels from Innoviris (Brussels Institute for Research and Innovation); S.K. is supported by a post-doctoral grant of the *Fonds Wetenschappelijk Onderzoek*, F.V.B. is *Maître de Recherches* of the *Fonds de la Recherche Scientifique*. This work was supported by the Interuniversity Attraction Poles Programme initiated by the Belgian Science Policy Office (programme IAP P7/28); the *Fonds de la Recherche Scientifique* (Grants 3.4.588.10F, 3.4530.12 and T.0134.13) and Innoviris. We thank Pr. Friedrich Götz (Universität Tübingen, Germany) for the kind gift of the ΔicaA mutant of the ATCC33591 strain, Pr. Gerald B. Pier (Brigham and Women's Hospital, Boston, MA), for the kind gift of the anti-PNAG antiserum, Tinne Buelens and Uwe Himmelreich (MoSAIC facility, KULeuven), for their help during bioluminescence imaging and Celia Lobo Romero and Cindy Colombo (KULeuven) for technical assistance during in vivo experimental procedures.

## Author contributions

W.S. performed the in vitro experiments; S.K., the in vivo experiments, A.B. and J.V., the electron microscopy. W.S., S.K. and F.V.B. designed the studies. P.V.D. and F.V.B. supervised the work with the help of M.-P.M.-L. and P.M.T. W.S., S.K. and F.V.B. wrote the manuscript. P.M.T., M.-P.M.-L. and P.V.D. contributed to the writing of the manuscript.
