## [Peer review file · Nature Communications]

Reviewers' comments:

Reviewer #1 (Remarks to the Author):

Comment on: "The antifungal caspofungin (Cas) increases moxifloxacin (Mox) activity against *Staphylococcus aureus* biofilms by inhibiting N-acetylglucosamine transferase activity"

The antifungal caspofungin inhibits the (1→3)-β-D-glucan synthesis and it has been thought that, based on protein similarity, it also might inhibit the β-1,6 N-acetyl-glucosamine transferase, IcaA; and indeed, it inhibited in vitro the IcaA transferase activity. It has been shown that caspofungin alone has little effect on *S. aureus* biofilms, however, in combination with the fluorquinolone, moxifloxacin, the viability of 24 h biofilm cells was decreased 1 to 6 logs, depending on strains. It has been shown that caspofungin dissolves the PIA structure in biofilms and it is thought that the small moxifloxacin molecule has better access to the cells. On the other hand the small linezolid or the larger molecules, vancomycin and daptomycin, showed no such synergistic effect.

Treatment of biofilm-associated infections is a real problem. It has turned out that once a biofilm was established, single antibiotic treatment normally fails. Therefore, combinations of antibiotics are increasingly used already to treat such infections. A systematic study was published in 2006 by Saginur et al. in AAC. Since then many more publications on improved treatment with combinations of antibiotics were published and some of the combinations are used already in therapy.

Bacterial viability in biofilms treated with Cas/Mox varied tremendously in the various strains. A reduction of only 75 % in one of the test strains means really little effect; although, with other the strains CFU was up to 7 logs decreased.

In animal tests it looks like as with Mox/Cas treatment after 6 days bacterial load is increasing again (Fig. 5).

Minor:

The Figs should be numbered

Fig. 1 is difficult to understand; what is CT for example?

Fig. 2 is also difficult to follow

References:

in 1, 12, 28 and many more - no journal is indicated - bacterial and gene names are not in italics

Reviewer #2 (Remarks to the Author):

Treating *Staphylococcus aureus* biofilm-related infections represent a huge unmet medical need. To this end, the submitted manuscript by Siala et al describes a combinatorial treatment where the use of the antifungal caspofungin appears to potentiate the antibiotic killing of moxifloxacin. They provide in vitro data that indicates that caspofungin inhibits the activity of *icaA*, a N-acetyl-glucosamine transferase, responsible for the synthesis of the exopolysaccharide PNAG. This impairment of catalytic activity results in changes to the biofilm architecture as determined through electron microscopy and the casp+mox treatment appears more potent at reducing *S. aureus* infection in a mouse model than a mox treatment alone. The results of this study are very interesting and the paper is generally well written, however this reviewer has several major concerns that need to be addressed before the manuscript could be accepted for publication. These are concerns and comments are outlined below.

Line 20/22: There are several references that are which are not up to date with the current understand of PNAG biosynthesis. For example, a citation is needed for the function of *icaB* (CE4

deacetylase) for example; Little et al. (JBC 2014) which provides both structural and functional information for this enzyme, and Atkin et al. (FEBS Lett. 2014) which suggests that icaC is involved O-succinylation of PNAG.

Line 39: The authors state that the aim of this study was to identify compounds that demonstrate synergy with antibiotics. However, none of this screening data is reported in this manuscript (as it should be) so it is difficult as a reviewer to understand how a moxifloxacin+caspofungin treatment was chosen. What are the results from this initial screen? How was this performed? It appears that the story could have developed with the inhibition of icaA by caspofungin.

Line 62: It is insufficient that bacterial viability is only measured using a CTC-based assay. Many cells in a biofilm can be metabolically dormant and therefore would not reduce the substrate. Thus, one could come to the erroneous conclusion that the antibiotic is more effective than is actually the case. Typically a CFU measurement is required to verify the number of viable bacteria following an antibiotic treatment. The authors use this later in the manuscript so why has this not been completed here? In my humble opinion this needs to be complete before one can believe this assay is valid. The authors need to report viability in terms of CFU rather than % reduction. Standard error in these measurements also needs to be included.

Line 69: How does treatment with an antibiotic result in a change in reducing biofilm biomass as detected using crystal violet? Can examples of this in the literature be used to justify this observation?

Line 79: It is not clear why these antibiotics were utilized instead of other fluoroquinolones such as ciprofloxacin or levofloxacin? The use of another fluoroquinolones would help the authors (Line 239) where the authors state the limitations of their work. If this combination treatment could function on other antibiotics in this class then this would be more supportive of their conclusions.

Line 143: Has calcofluor white staining been previously used to stain PNAG biofilms? If so, could a reference be provided? This stain is specific for beta 1,3 and beta 1,4 linkages, and PNAG is beta 1,6 linked. There does not appear to be a PNAG knockout control that is used to demonstrate that this stain directly targets PNAG.

Line 148: The authors use DLS to measure the size of the purified polysaccharide. Typically the "size" of polysaccharides is measured by assaying for the length of the polysaccharide. The authors should report PNAG length in terms of the degree of polymerization (DP), which is a ratio of glucosamine to reducing ends. Assaying for the degree of acetylation could dramatically help the paper and may provide insight as to the mechanism of action of antibiotic potentiation. One major limitation of using DLS is that the particle size is influenced by parameters such as aggregation that in turn are the result of the biochemical properties of the polymer. Inhibition of icaA has implications downstream as this enzyme provides the substrate for icaB and icaC. It is well established in the literature that changes in deacetylation change the properties of PNAG as the polymer becomes either more or less cationic.

Line 238: The authors state that the large molecular weight of some antibiotics may prevent them from working more effectively in the presence of caspofungin. However they fail to mention that other alterations such as deacetylation, O-succinylation could also dramatically effect the rate of penetration of these antibiotics (see comment for Line 148). The molecular weight does not seem to be a solid argument since the authors stated on Line 131 that caspofungin treatment did not equally increase penetration of mox in their four tested strains. Therefore, there must be another mechanism at play. Perhaps the authors could mention the contributions of protein and eDNA in the biofilm which are still present in the absence of exopolysaccharides.

Figure 1: This figure needs to be explained in the caption in more detail. For example what is the difference between the black boxes vs the white triangles.

Reviewer #1

1. Treatment of biofilm-associated infections is a real problem. It has turned out that once a biofilm was established, single antibiotic treatment normally fails. Therefore, combinations of antibiotics are increasingly used already to treat such infections. A systematic study was published in 2006 by Saginur et al. in AAC. Since then many more publications on improved treatment with combinations of antibiotics were published and some of the combinations are used already in therapy.

REPLY: It is unclear to us whether this sentence is only a general comment or a request to elaborate on the interest of drug combinations against biofilms. In the introduction of our original submission, we focused our interest on combinations between antibiotics and non-antibiotic agents disrupting the matrix, which is the topic of our paper. We entirely agree with the reviewer that antibiotic combination is a strategy which has been explored over the last years to try eradicating biofilms. This is actually the motto of our paper.

ACTION TAKEN: we have added a sentence about the use of antibiotic combinations to act against biofilms in our Introduction, and have referred to the paper of Sagninur *et al.* suggested by the referee. See changes marked **A.1.** in the revised version.

2. Bacterial viability in biofilms treated with Cas/Mox varied tremendously in the various strains. A reduction of only 75 % in one of the test strains means really little effect; although, with other the strains CFU was up to 7 logs decreased.

REPLY: we wish here to point to the attention of the reviewer that we measured a reduction in the fluorescence signal of resorufin (expressed in percentage of control values), and not a percentage of reduction in viable counts. Thus, a reduction of 75 % of this signal is not equivalent to a reduction of 75% in viability. We actually deal here with a 4 log CFU reduction, which is a major effect. This stems from additional experiments requested by Reviewer #2. This reviewer specifically requests that we examine the correlation between the reduction in fluorescence signal and in bacterial counts, which we now have done. Please see our reply to Reviewer #2 for more details and the data presented in the new Figure 1 (and the new supplementary Figure 2 for other strains).

ACTION TAKEN: see the modifications made in reply to comment #3 of Reviewer #2.

3. In animal tests it looks like as with Mox/Cas treatment after 6 days bacterial load is increasing again (Fig. 5).

REPLY: the reviewer is probably alluding to the presence of a small red spot re-appearing at day 6 in the treated mice.

As illustrated in the figure below which shows individual measures over time for mice treated by the combination moxifloxacin + caspofungin, there is no major difference in the signal measured between day 5, day 6, and day 7. We therefore do not think this corresponds to a regrowth of the bacteria, but rather to inter-individual variability among mice, which seems visible on pictures but do not translate in a significant change in the signal intensity.

ACTION TAKEN: we have modified the figure illustrating BLI data by adding panels showing individual counts for mice treated by fluoroquinolones alone vs fluoroquinolones combined with caspofungin. We also provide a picture of the mice treated by the combination at day 7. See change marked **A.3.** in the revised version.

Minor:

4. The Figs should be numbered

REPLY: We apologize if the figures were not numbered. Their number was included in the file name and also introduced on the website upon submission, but it does not seem to be re-transcribed on the figures themselves.

ACTION TAKEN: we have now added the number of the figures in their upper left corner.

5. Fig. 1 is difficult to understand; what is CT for example?

REPLY: We apologize if our legend was not clear enough. CT means control (i.e. biofilm not exposed to fluoroquinolones or caspofungin). CV (triangles) means crystal violet and RF (squares for moxifloxacin/circles for delafloxacin in the revised version) means resorufin fluorescence.

ACTION TAKEN: We have modified the caption of the figure, hoping is it now easier to understand. See changes marked **A.5.** in the revised version (this figure has now been moved to supplemental material).

6. Fig. 2 is also difficult to follow

REPLY: Again, we apologize if the figure was difficult to understand. The way we presented the data is the same as we used in a previous publication (Siala *et al.*, 2014). This mode of representation allows to easily show the increase in activity upon caspofungin addition for each individual strain in a synoptic fashion.

ACTION TAKEN: We have modified the caption of the figure, trying to make it more explicit with respect on the way the figure has to be interpreted. We hope these changes will meet the reviewer's concerns. We also briefly explain how the figure needs to be interpreted in the text. See changes marked **A.6** in the revised version.

7. References:

in 1, 12 , 28 and many more - no journal is indicated - bacterial and gene names are not in italics

REPLY: The reviewer is right. We used Reference Manager version 12 to generate the reference list, and this software does not support italics in titles. There was also a mistake in the format used, which is why some journal names did not appear. We apologize for the difficulty it may have created when reviewing the paper.

ACTION TAKEN: We have checked that the references do now follow the format of the Journal. We manually put in italics bacteria and gene names.

Reviewer #2:

1. Line 20/22: There are several references that are which are not up to date with the current understand of PNAG biosynthesis. For example, a citation is needed for the function of *icaB* (CE4 deacetylase) for example; Little *et al.* (JBC 2014) which provides both structural and functional information for this enzyme, and Atkin *et al.* (FEBS let. 2014) which suggests that *icaC* is involved O-succinylation of PNAG.

REPLY: We thank the reviewer for this remark, which is right. In our original submission, we focused on *princeps* papers that describe the function of these enzymes.

ACTION TAKEN: We have now re-examined in details the recent literature on the topic and have reviewed our text accordingly. We have also included the references suggested by the reviewer. See changes marked **B.1** in the revised version.

2. Line 39: The authors state that the aim of this study was to identify compounds that demonstrate synergy with antibiotics. However, none of this screening data is reported in this manuscript (as it should be) so it is difficult as a reviewer to understand how a moxifloxacin+caspofungin treatment was chosen. What are the results from this initial screen?

How was this performed? It appears that the story could have developed with the inhibition of *icaA* by caspofungin.

REPLY:

Our screening approach was in two steps:

First, we looked for compounds (to be combined with moxifloxacin) having an amphipathic character but that did not show activity against *S. aureus* by themselves, in order to identify compounds showing a synergy towards biofilms by an effect independent of their antimicrobial activity. The rationale of our reasoning was that detergents are quite active on biofilms and are often amphipathic. Thus, our hypothesis was that amphiphilicity could help penetration and deconstruction of the matrix.

Next, we restricted our analysis to compounds that are already used in the clinics, as this may facilitate clinical applications since these compounds are already approved for human use.

We also think that reporting the way we designed our screening in the paper will rather be confusing, because, at the end of the day, the mechanism by which caspofungin improves fluoroquinolone activity is unrelated to its amphipathic character. This is the way science makes progress: testing hypotheses and then revisiting the initial model based on unexpected results.

We, however, understand that the reader may have been frustrated when reading the introduction as it was written in our original submission.

ACTION TAKEN:

We reworded our text in order to present caspofungin as a rational selection based on the fact it is known to inhibit glycan synthesis in fungi. See changes marked **B.2** in the revised version. We hope this modification makes the introduction easier to read and more in line with the content of the paper.

3. Line 62: It is insufficient that bacterial viability is only measured using a CTC-based assay. Many cells in a biofilm can be metabolically dormant and therefore would not reduce the substrate. Thus, one could come to the erroneous conclusion that the antibiotic is more effective than is actually the case. Typically a CFU measurement is required to verify the number of viable bacteria following an antibiotic treatment. The authors use this later in the manuscript so why has this not been completed here? In my humble opinion this needs to be complete before one can believe this assay is valid. The authors need to report viability in terms of CFU rather than % reduction. Standard error in these measurements also needs to be included.

REPLY: We thank the reviewer for this useful suggestion. Actually, a similar issue was also raised by Referee #1 (see comment #2 from Reviewer #1).

The correlation between resorufin fluorescence and CFU has been presented in previous publications (Pettit *et al.*, 2005; Pettit *et al.*, 2009; Tawakoli *et al.*, 2015; Van den Driessche *et al.*, 2014; van der Waal *et al.*, 2012), including by our group (Bauer *et al.*, 2013).

The advantage of the resazurin assay is that it can be applied in parallel for several conditions, explaining why it is preferred for high throughput screening experiments (Pettit *et al.*, 2009; van der Waal *et al.*, 2012) or for performing full concentrations-effect relationship needed for evaluation of pharmacodynamic parameters, like antibiotic potency or maximal efficacy (Bauer *et al.*, 2013; Siala *et al.*, 2014; Vandeveldel *et al.*, 2015). This is why we used it in the present study.

We however do agree that it remains an indirect measure of antibiotic activity as (a) it does not allow distinguishing between bacteriostatic and bactericidal activity of antibiotics (as acknowledged by ourselves in a poster presented at the 24th European Congress of Clinical Microbiology and Infectious Diseases (ECCMID), Barcelona, Spain, 2014 :

<http://www.facm.ucl.ac.be/posters/2014/ECCMID-2014/Vandeveldel-ECCMID-2014-NADH-oxidase-eP288.pdf>) and (b) it does not give a direct assessment of the extent of bacterial killing.

ACTION TAKEN: we have redone all experiments presented in Figure 1 from our original submission for all the strains used in the paper and assessed the activity of treatments by measuring the changes in CFU. We observed that CFU counts and fluorescence signal are well correlated throughout the whole data set.

These data have now been included in the paper as a main figure. See new Figure 1 and changes in the corresponding text (methods and results) marked **B.3**. The former Figure 1 has now been moved to the supplemental material as a support to Figure 2 (full dose response used to calculate potency) and also because it shows the effect of treatments on biomass.

4. Line 69: How does treatment with an antibiotic result in a change in reducing biofilm biomass as detected using crystal violet? Can examples of this in the literature be used to justify this observation?

REPLY: this is again an interesting question, but already discussed in one of our previous publications (Bauer *et al.*, 2013): We wrote in the discussion of this paper: “*We see that the reduction in antibiotic activity with biofilm maturation seems more important with respect to biofilm mass than to bacterial viability. This is consistent with the fact that antibacterial agents act on essential bacterial targets and not upon biofilm matrix. Reductions in biofilm mass are thus likely consecutive to bacterial growth inhibition or killing during the 48 h of exposure to antibiotics, as previously demonstrated for daptomycin or fluoroquinolones*”.

A series of publications by other groups documents a reduction in biomass upon biofilm exposure to high antibiotic concentrations, including fluoroquinolones (Gattringer *et al.*, 2010; Mu *et al.*, 2016; Nguyen *et al.*, 2016; Presterl *et al.*, 2005; Rojo-Molinero *et al.*, 2016; Roveta *et al.*, 2008; Roveta *et al.*, 2007; Skogman *et al.*, 2012; Tetz *et al.*, 2009; Thangamani *et al.*, 2015).

Crystal violet staining has even been used as a high throughput method for identifying compounds able to disrupt biofilms, including antibiotics like colistin, fluoroquinolones, β -lactams, or rifampicin (Sandberg *et al.*, 2008; Skogman *et al.*, 2012).

None of these publications, however, provides a clear mechanistic support to these observations. It has simply been proposed that bacterial killing can lead to a deconstruction of the biofilm ([Rojo-Moliner *et al.*, 2016]; see also our own paper cited here above [Bauer *et al.*, 2013]). This hypothesis is coherent with the observation that bacterial death within biofilms is associated with a facilitation of biofilm dispersal, related to the creation of voids within the matrix (Kostakioti *et al.*, 2013; Webb *et al.*, 2003).

Moreover, it has also been suggested that cell wall-acting antibiotics (β -lactams or glycopeptides) can increase *agr* expression (Kavanaugh and Horswill, 2016) including in biofilms (Tan *et al.*, 2015), stimulating thereby biofilm dispersion (Boles and Horswill, 2008; Mirani *et al.*, 2013). At this stage, however, there is no evidence that this type of mechanism may take place in biofilms treated by fluoroquinolones.

ACTION TAKEN: The original Figure 1 has been moved to the supplemental material (as part of our reply to comment #3), which put less emphasis on the effect of antibiotics on biomass. Nevertheless, we have now commented in the discussion on the potential link between bacterial killing and biofilm dispersal, which may explain the reduction in biomass as observed using crystal violet staining. See change marked **B.4** in the revised version.

5. Line 79: It is not clear why these antibiotics were utilized instead of other fluoroquinolones such as ciprofloxacin or levofloxacin? The use of another fluoroquinolones would help the authors (Line 239) where the authors state the limitations of their work. If this combination treatment could function on other antibiotics in this class then this would be more supportive of their conclusions.

REPLY: we used moxifloxacin as a fluoroquinolone rather than ciprofloxacin or levofloxacin simply because moxifloxacin is so far the more potent fluoroquinolone on Gram-positive bacteria among those approved for clinical use and readily available for clinicians (Jones *et al.*, 1999; Schmitz *et al.*, 1998; Van Bambeke *et al.*, 2005). Ciprofloxacin is definitely not recommended for infections caused by Gram-positive bacteria and levofloxacin is less potent than moxifloxacin. Thus, we did not wish to indirectly promote the use of these antibiotics for staphylococcal infections.

We however understand the concern of the referee and do agree that extending the observation to another fluoroquinolone targeted against Gram-positive bacteria would be useful. We happened to have access to a new fluoroquinolone, delafloxacin, with increased activity against Gram-positive bacteria. We previously showed that delafloxacin it is active also against staphylococcal biofilms (Siala *et al.*, 2014).

ACTION TAKEN: We repeated all experiments (both *in vitro* and *in vivo*) made with moxifloxacin using now delafloxacin, a fluoroquinolone currently in phase III clinical trials (Van Bambeke,

2015). Delafloxacin eventually proved more potent than moxifloxacin against Gram-positive bacteria in general and *S. aureus* in particular (Harnett *et al.*, 2004; Lemaire *et al.*, 2011).

The new set of data we obtained leads us to conclude that the data obtained with moxifloxacin can be extended to other fluoroquinolones directed against Gram-positive bacteria. We eventually found that delafloxacin is globally more potent than moxifloxacin when used alone. We also observed a synergy between delafloxacin and caspofungin, which is best seen against strains expressing *icaA* at a high level.

Thus, our results with delafloxacin have been added in the set of data submitted for publication.

See changes introduced along the paper in figures and tables, as well as the text (marked as **B.5.**)

6. Line 143: Has calcofluor white staining been previously used to stain PNAG biofilms? If so, could a reference be provided? This stain is specific for beta 1,3 and beta 1,4 linkages, and PNAG is beta 1,6 linked. There does not appear to be a PNAG knockout control that is used to demonstrate that this stain directly targets PNAG.

REPLY: the reviewer is right, calcofluor white is not specific for PNAG (poly- β (1-6)-N-acetylglucosamine), as it preferentially binds to sugars with β -1,3 and β -1,4 linkages (Rasconi *et al.*, 2009). Nevertheless, it has already been used in the literature to determine the levels of 1,6- β -glucans in fungal cell walls (Abeijon and Chen, 1998; Ram and Klis, 2006; Vink *et al.*, 2002).

We do not have a knock out strain for PNAG production. We therefore undertook to specifically detect PNAG within the biofilms using an antibody (dot-blot analysis). We observed that spot intensity in these dot blots was markedly reduced in biofilms exposed to caspofungin, except in strain 2003/651 which is resistant to caspofungin effects.

ACTION TAKEN: the data using calcofluor white were removed. Immunoblot of PNAG are presented instead. See new Figure 8, and the corresponding changes made in the methods and results section (marked **B.6.** in the revised version).

7. Line 148: The authors use DLS to measure the size of the purified polysaccharide. Typically the "size" of polysaccharides is measured by assaying for the length of the polysaccharide. The authors should report PNAG length in terms of the degree of polymerization (DP), which is a ratio of glucosamine to reducing ends. Assaying for the degree of acetylation could dramatically help the paper and may provide insight as to the mechanism of action of antibiotic potentiation. One major limitations of using DLS is that the particle size is influenced by parameters such as aggregation that in turn are the result of the biochemical properties of the polymer. Inhibition of *icaA* has implications downstream as this enzyme provides the substrate for *icaB* and *icaC*. It is well established in the literature that changes in deacetylation change the properties of PNAG as the polymer becomes either more or less cationic.

REPLY: We thank the reviewer for this interesting remark. We took it into account and used a highly sensitive method in order to determine PNAG length. PNAG were digested by Dispersin B (N-acetyl- β -hexosaminidase) that cleaves the β -1,6-linkages and release monomers of N-acetylglucosamine [GlcNAc] (Itoh *et al.*, 2005). We then quantified GlcNAc ends using the Morgan-Elson reaction (Elson and Morgan, 1933) coupled to a fluorimetric detection (Takahashi *et al.*, 2003). Thus, the intensity of the signal will be proportional to the initial degree of polymerization, the method being based on the detection of newly generated GlcNAc termini (Takahashi *et al.*, 2003).

By using this approach, we were able to demonstrate (a) that the concentration of GlcNAc liberated by dispersin B was proportional to the level of *icaA* expression for each individual strain and (b) that this concentration was markedly reduced in caspofungin-treated biofilms, except in strain 2003/651 which was resistant to caspofungin effects. Thus, the experiment suggested by the reviewer is strongly supporting the hypothesis that caspofungin acts by impairing PNAG polymerization.

We therefore warmly thank the reviewer for this useful suggestion.

ACTION TAKEN: DLS data have been removed from the paper; GlcNAc quantification in control and caspofungin-exposed biofilms were included instead. See new Figure 8 and changes marked **B.7** in the revised version.

8. Line 238: The authors state that the large molecular weight of some antibiotics may prevent them from working more effectively in the presence of caspofungin. However they fail to mention that other alterations such as deacetylation, O-succinylation could also dramatically affect the rate of penetration of these antibiotics (see comment for Line 148). The molecular weight does not seem to be a solid argument since the authors stated on Line 131 that caspofungin treatment did not equally increase penetration of mox in their four tested strains. Therefore, there must be another mechanism at play. Perhaps the authors could mention the contributions of protein and eDNA in the biofilm which are still present in the absence of exopolysaccharides.

REPLY: We do agree with the reviewer that the molecular weight of the antibiotic is not the only determinant explaining why their activity in biofilms is not increased in the presence of caspofungin. Biofilm matrix is a complex structure, and antibiotic penetration most probably depends on antibiotics biophysical properties, as well as of matrix physicochemical properties. Evaluating the interaction of each antibiotic class with individual matrix constituents deserves further studies, but which are clearly out of the scope of the present work, which is rather focused on the fluoroquinolone-caspofungin combination.

Regarding succinylation, this is indeed an interesting hypothesis. We can simply argue at this stage that we did not observe significant differences in the level of expression of *icaC* among strains (see the data presented in supplementary Table 2).

Yet, to take into account the remark of the reviewer, we have examined the concentration of proteins and DNA in the matrix of biofilms. We noticed differences among strains with respect

to the matrix content in proteins but did not observe any effect of caspofungin on the biofilm content in proteins or in eDNA. This add another argument pleading for the specificity of action of caspofungin.

ACTION TAKEN: we have added the data on protein and eDNA content in our revised manuscript.

As requested by the reviewer, we have amended our discussion in order to suggest a possible role of other matrix constituents in limiting the activity of other antibiotic classes. See changes marked **B.8**. in the revised manuscript.

9. Figure 1: This figure needs to be explained in the caption in more detail. For example what is the difference between the black boxes vs the white triangles.

REPLY: we apologize if the caption of this figure was not clear enough. The symbols were actually described at the top of the figure.

ACTION TAKEN: the caption of the figure now mentions also an explanation of each symbol. See change marked **B.9**. in the revised version.

References to this reply (they are not systematically included in the paper)

- Abeijon C and Chen L Y (1998) The role of glucosidase I (Cwh41p) in the biosynthesis of cell wall beta-1,6-glucan is indirect. *Mol Biol Cell* **9**: 2729-2738.
- Bauer J, Siala W, Tulkens P M and Van Bambeke F (2013) A combined pharmacodynamic quantitative and qualitative model reveals the potent activity of daptomycin and delafloxacin against *Staphylococcus aureus* biofilms. *Antimicrob Agents Chemother* **57**: 2726-2737.
- Boles BR and Horswill A R (2008) Agr-mediated dispersal of *Staphylococcus aureus* biofilms. *PLoS Pathog* **4**: e1000052.
- Elson LA and Morgan W T (1933) A colorimetric method for the determination of glucosamine and chondrosamine. *Biochem J* **27**: 1824-1828.
- Gattringer KB, Suchomel M, Eder M, Lassnigg A M, Graninger W and Presterl E (2010) Time-dependent effects of rifampicin on staphylococcal biofilms. *Int J Artif Organs* **33**: 621-626.
- Harnett SJ, Fraise A P, Andrews J M, Jevons G, Brenwald N P and Wise R (2004) Comparative study of the in vitro activity of a new fluoroquinolone, ABT-492. *J Antimicrob Chemother* **53**: 783-792.
- Itoh Y, Wang X, Hinnebusch B J, Preston J F and Romeo T (2005) Depolymerization of beta-1,6-N-acetyl-D-glucosamine disrupts the integrity of diverse bacterial biofilms. *J Bacteriol* **187**: 382-387.
- Jones ME, Visser M R, Klootwijk M, Heisig P, Verhoef J and Schmitz F J (1999) Comparative activities of clinafloxacin, grepafloxacin, levofloxacin, moxifloxacin, ofloxacin, sparfloxacin, and trovafloxacin and nonquinolones linozolid, quinupristin-dalfopristin, gentamicin, and vancomycin against clinical isolates of ciprofloxacin-resistant and -susceptible *Staphylococcus aureus* strains. *Antimicrob Agents Chemother* **43**: 421-423.
- Kavanaugh JS and Horswill A R (2016) Impact of Environmental Cues on Staphylococcal Quorum Sensing and Biofilm Development. *J Biol Chem* **291**: 12556-12564.
- Kolodkin-Gal I, Cao S, Chai L, Bottcher T, Kolter R, Clardy J and Losick R (2012) A self-produced trigger for biofilm disassembly that targets exopolysaccharide. *Cell* **149**: 684-692.
- Kostakioti M, Hadjifrangiskou M and Hultgren S J (2013) Bacterial biofilms: development, dispersal, and therapeutic strategies in the dawn of the postantibiotic era. *Cold Spring Harb Perspect Med* **3**: a010306.
- Lemaire S, Tulkens P M and Van Bambeke F (2011) Contrasting effects of acidic pH on the extracellular and intracellular activities of the anti-gram-positive fluoroquinolones moxifloxacin and delafloxacin against *Staphylococcus aureus*. *Antimicrob Agents Chemother* **55**: 649-658.
- Mirani ZA, Aziz M, Khan M N, Lal I, Hassan N U and Khan S I (2013) Biofilm formation and dispersal of *Staphylococcus aureus* under the influence of oxacillin. *Microb Pathog* **61-62**: 66-72.
- Mu H, Tang J, Liu Q, Sun C, Wang T and Duan J (2016) Potent Antibacterial Nanoparticles against Biofilm and Intracellular Bacteria. *Sci Rep* **6**: 18877.
- Nguyen TK, Selvanayagam S, Ho K K, Chen R, Kutty S K, Rice S A, Kumar N, Barraud N, Huong H T T and Boyer C (2016) Co-delivery of nitric oxide and antibiotic using polymeric nanoparticles. *Chem Sci* **7**: 1016-1027.
- Pettit RK, Weber C A, Kean M J, Hoffmann H, Pettit G R, Tan R, Franks K S and Horton M L (2005) Microplate Alamar blue assay for *Staphylococcus epidermidis* biofilm susceptibility testing. *Antimicrob Agents Chemother* **49**: 2612-2617.
- Pettit RK, Weber C A and Pettit G R (2009) Application of a high throughput Alamar blue biofilm susceptibility assay to *Staphylococcus aureus* biofilms. *Ann Clin Microbiol Antimicrob* **8**: 28.
-

- Presterl E, Grisold A J, Reichmann S, Hirschl A M, Georgopoulos A and Graninger W (2005) Viridans streptococci in endocarditis and neutropenic sepsis: biofilm formation and effects of antibiotics. *J Antimicrob Chemother* **55**: 45-50.
- Ram AFJ and Klis F M (2006) Identification of fungal cell wall mutants using susceptibility assays based on Calcofluor white and Congo red. *Nat Protoc* **1**: 2253-2256.
- Rasconi S, Jobard M, Jouve L and Sime-Ngando T (2009) Use of calcofluor white for detection, identification, and quantification of phytoplanktonic fungal parasites. *Appl Environ Microbiol* **75**: 2545-2553.
- Rojo-Molinero E, Macia M D, Rubio R, Moya B, Cabot G, Lopez-Causape C, Perez J L, Canton R and Oliver A (2016) Sequential Treatment of Biofilms with Aztreonam and Tobramycin Is a Novel Strategy for Combating *Pseudomonas aeruginosa* Chronic Respiratory Infections. *Antimicrob Agents Chemother* **60**: 2912-2922.
- Roveta S, Marchese A and Schito G C (2008) Activity of daptomycin on biofilms produced on a plastic support by *Staphylococcus* spp. *Int J Antimicrob Agents* **31**: 321-328.
- Roveta S, Schito A M, Marchese A and Schito G C (2007) Activity of moxifloxacin on biofilms produced in vitro by bacterial pathogens involved in acute exacerbations of chronic bronchitis. *Int J Antimicrob Agents* **30**: 415-421.
- Sandberg M, Maattanen A, Peltonen J, Vuorela P M and Fallarero A (2008) Automating a 96-well microtitre plate model for *Staphylococcus aureus* biofilms: an approach to screening of natural antimicrobial compounds. *Int J Antimicrob Agents* **32**: 233-240.
- Schmitz FJ, Hofmann B, Hansen B, Scheuring S, Luckefahr M, Klootwijk M, Verhoef J, Fluit A, Heinz H P, Kohrer K and Jones M E (1998) Relationship between ciprofloxacin, ofloxacin, levofloxacin, sparfloxacin and moxifloxacin (BAY 12-8039) MICs and mutations in *grlA*, *grlB*, *gyrA* and *gyrB* in 116 unrelated clinical isolates of *Staphylococcus aureus*. *J Antimicrob Chemother* **41**: 481-484.
- Siala W, Mingeot-Leclercq M P, Tulkens P M, Hallin M, Denis O and Van Bambeke F (2014) Comparison of the antibiotic activities of Daptomycin, Vancomycin, and the investigational Fluoroquinolone Delafloxacin against biofilms from *Staphylococcus aureus* clinical isolates. *Antimicrob Agents Chemother* **58**: 6385-6397.
- Skogman ME, Vuorela P M and Fallarero A (2012) Combining biofilm matrix measurements with biomass and viability assays in susceptibility assessments of antimicrobials against *Staphylococcus aureus* biofilms. *J Antibiot (Tokyo)* **65**: 453-459.
- Takahashi T, Ikegami-Kawai M, Okuda R and Suzuki K (2003) A fluorimetric Morgan-Elson assay method for hyaluronidase activity. *Anal Biochem* **322**: 257-263.
- Tan X, Qin N, Wu C, Sheng J, Yang R, Zheng B, Ma Z, Liu L, Peng X and Jia A (2015) Transcriptome analysis of the biofilm formed by methicillin-susceptible *Staphylococcus aureus*. *Sci Rep* **5**: 11997.
- Tawakoli PN, Sauer B, Becker K, Buchalla W and Attin T (2015) Interproximal biofilm removal by intervallic use of a sonic toothbrush compared to an oral irrigation system. *BMC Oral Health* **15**: 91.
- Tetz GV, Artemenko N K and Tetz V V (2009) Effect of DNase and antibiotics on biofilm characteristics. *Antimicrob Agents Chemother* **53**: 1204-1209.
- Thangamani S, Younis W and Seleem M N (2015) Repurposing ebselen for treatment of multidrug-resistant staphylococcal infections. *Sci Rep* **5**: 11596.
- Van Bambeke F, Michot J M, Van Eldere J and Tulkens P M (2005) Quinolones in 2005: an update. *Clin Microbiol Infect* **11**: 256-280.
- Van Bambeke F (2015) Delafloxacin, a non-zwitterionic fluoroquinolone in Phase III of clinical development: evaluation of its pharmacology, pharmacokinetics, pharmacodynamics and clinical efficacy. *Future Microbiol* **10**: 1111-1123.
-

- Van den Driessche F, Rigole P, Brackman G and Coenye T (2014) Optimization of resazurin-based viability staining for quantification of microbial biofilms. *J Microbiol Methods* **98**: 31-34.
- van der Waal SV, Jiang L M, de Soet J J, van der Sluis L W M, Wesselink P R and Crielaard W (2012) Sodium chloride and potassium sorbate: a synergistic combination against *Enterococcus faecalis* biofilms: an in vitro study. *Eur J Oral Sci* **120**: 452-457.
- Vandevelde NM, Tulkens P M, Muccioli G G and Van Bambeke F (2015) Modulation of the activity of moxifloxacin and solithromycin in an in vitro pharmacodynamic model of *Streptococcus pneumoniae* naive and induced biofilms. *J Antimicrob Chemother* **70**: 1713-1726.
- Vink E, Vossen J H, Ram A F J, van den Ende H, Brekelmans S, de Nobel H and Klis F M (2002) The protein kinase Kic1 affects 1,6-beta-glucan levels in the cell wall of *Saccharomyces cerevisiae*. *Microbiology* **148**: 4035-4048.
- Webb JS, Thompson L S, James S, Charlton T, Tolker-Nielsen T, Koch B, Givskov M and Kjelleberg S (2003) Cell death in *Pseudomonas aeruginosa* biofilm development. *J Bacteriol* **185**: 4585-4592.
-

REVIEWERS' COMMENTS:

Reviewer #1 (Remarks to the Author):

All the points raised by reviewer 1 were satisfactorily addressed.
In ref. 13 and 17 it should be 'Götz'

Reviewer #2 (Remarks to the Author):

The authors have completed a number of additional experiments based upon the recommendations of both reviewers. This new data includes testing additional antibiotics, correlating CFU with a metabolic assay, and more accurately characterizing the PNAG polysaccharide under experimental conditions. This reviewer is impressed as the resulting manuscript is significantly improved with the inclusion of this new work. The conclusions that are drawn are more impactful and supportive of the new data. Additionally, the figures are better over the previous version and the references have been updated to include the latest work in the field. I believe that these substantial improvements merit publication in the journal.